METHODS AND RESOURCES

# Human striatal organoids derived from pluripotent stem cells recapitulate striatal development and compartments

Xinyu Chen[1☯], Hexige Saiyin[2☯], Yang Liu[1☯], Yuqi Wang[2], Xuan Li[3], Rong Ji[4], Lixiang Ma[1]*

1 Department of Anatomy and Histology & Embryology, School of Basic Medical Sciences, Fudan University, Shanghai, P.R. China, 2 State Key Laboratory of Genetic Engineering, School of Life Sciences, Fudan University, Shanghai, P.R. China, 3 The Fifth Affiliated Hospital Sun Yat-Sen University, Zhuhai, P.R. China, 4 Department of Neurology, Huadong Hospital, Fudan University, Shanghai, P.R. China

☯ These authors contributed equally to this work.
* lxma@fudan.edu.cn

**Data Availability Statement:** The sequencing datasets generated and/or analyzed during this study are available in the Gene Expression Omnibus with the following accession numbers for

## Abstract

The striatum links neuronal circuits in the human brain, and its malfunction causes neuronal disorders such as Huntington's disease (HD). A human striatum model that recapitulates fetal striatal development is vital to decoding the pathogenesis of striatum-related neurological disorders and developing therapeutic strategies. Here, we developed a method to construct human striatal organoids (hStrOs) from human pluripotent stem cells (hPSCs), including hStrOs-derived assembloids. Our hStrOs partially replicated the fetal striatum and formed striosome and matrix-like compartments in vitro. Single-cell RNA sequencing revealed distinct striatal lineages in hStrOs, diverging from dorsal forebrain fate. Using hStrOs-derived assembloids, we replicated the striatal targeting projections from different brain parts. Furthermore, hStrOs can serve as hosts for striatal neuronal allografts to test allograft neuronal survival and functional integration. Our hStrOs are suitable for studying striatal development and related disorders, characterizing the neural circuitry between different brain regions, and testing therapeutic strategies.

## Introduction

Pluripotent stem cell-based central nervous system (CNS) models have rapidly evolved since the advent of neural rosette formation from human embryonic stem cells (hESCs) [1]. Techniques to derive a targeted neuron from human pluripotent stem cells (hPSCs) are sophisticated [2–5]. Based on 2-dimensional (2D) differentiation methods, various protocols have been published to construct 3-dimensional (3D) brain organoids through guided and unguided methods [6–9], which are self-organized and partially recapitulate the neuronal activities of the human brain. Brain organoid technology provides unique opportunities to anatomically and spatially characterize the development of the human brain. The flexibility of organoids provides more chances to study complex neural systems; for example, fused organoids have been used to describe neuronal projections and migration between multiple brain regions [9–11].

readers: GSE183903. Raw data are within the paper and its Supporting Information files: S1 Data and S2 Data.

**Funding:** This work was supported by the National Natural Science Foundation of China (82071269 to LM), the National Key Research and Development Program of China (2018YFA0108004 and 2021YFA1101302 to LM), and the Shanghai Municipal Planning Commission of Science and Research Fund (201840009 to LM). The funders had no role in study design, data collection and analysis, decision to publish, or preparation of the manuscript.

**Competing interests:** The authors have declared that no competing interests exist.

**Abbreviations:** CNS, central nervous system; GE, ganglionic eminence; GFP, green fluorescent protein; hCO, human cortical organoid; HD, Huntington's disease; hESC, human embryonic stem cell; hiPSC, human-induced pluripotent stem cell; hMO, human midbrain organoid; hPSC, human pluripotent stem cell; hStrO, human striatal organoid; LGE, lateral ganglionic eminence; MGE, medial ganglionic eminence; MSN, medium spiny neuron; MZ, mantle zone; PCA, principal component analysis; PCW, postconceptional week; scRNA-seq, single-cell RNA-sequencing; SVZ, subventricular zone; UMAP, uniform manifold approximation and projection; UMI, unique molecular identifier; VZ, ventricular zone.

The striatum, the gateway of the basal ganglia, receives inputs from the cerebral cortex and the thalamus while forming a complicated projection relationship with the substantia nigra and the pallidum in the midbrain [12,13]. It develops from the lateral ganglionic eminences (LGEs) located ventral to the developing forebrain [14]. The mature striatum contains 95% medium spiny neurons (MSNs) [15]. Striatal development is a complex process followed by 2 distinct yet complimentary basic, organizational programs [15]. The programming generates 2 compartments with different neurochemical signatures refer to as striosome and matrix, which fully intermingled the direct pathway MSNs and indirect pathway MSNs [15]. Recently, a method to generate an organoid resembling the striatum has been reported [16]. To date, human corticogenesis has been well characterized in human cortical organoids (hCOs), [17] whereas the similar developmental dynamics in hStrOs are not well illustrated; consequently, this lack of understanding limits its application in disease modeling and targeted manipulation. Therefore, detailed and systemic phenotyping of developing organoids is necessary to enhance the repertoire of phenotypic assays available for hStrOs.

Here, we describe a 3D culturing system to generate human brain organoids resembling the striatum using facile techniques. We characterized the development of hStrOs by enhancing phenotypic analyses. The data from single-cell RNA-sequencing (scRNA-seq) of Day 110 hStrOs also showed similar cellular organizations and developmental trajectories similar to those of the developing human striatum. Mainly, we reported a self-organized regionalization in developing hStrO, which was similar to the compartmentalization in developing striatum. Moreover, we have expanded the potential applications of hStrOs, including using fused organoids to reconstruct the projection target striatum and disease modeling in vitro.

## Results

### Generation of human striatal organoids

We used hESC-H9, an hESC, and hiPSC-8-12, a human-induced pluripotent stem cell (hiPSC) derived from a healthy female at age 19 to generate hStrOs by modifying the cortical organoid approach and referring to the MSN neuron induction protocol [4,8]. Based on our previous 2D culture protocol of MSNs, 0.65 μm purmorphamine (Pur), an SHH pathway agonist, was used to induce LGE fates [4]. We further tested different dosages of Pur (0, 0.7, and 0.75 μm) to determine the dose-dependent LGE fates in 3D cultures (**Fig 1A and 1B**). All organoids expanded when maintained in spinning culture (**Fig 1B**). By Day 45, we observed that 78.81 ± 1.997% of cells dissociated co-expressed DARPP32 and MAP2, which represent MSNs in the striatum, at a Pur concentration of 0.65 μm (**Fig 1C and 1D**). The proportion was significantly higher than that in the groups treated with the Pur concentrations of 0 μm (15.41 ± 2.012%, $P < 0.001$), 0.7 μm (57.35 ± 2.289%, $P < 0.001$), and 0.75 μm (41.79 ± 2.557%, $P < 0.001$), suggesting that Pur-treated organoids underwent efficient LGE development (**Fig 1C and 1D**).

We further analyzed the tissues in vitro in terms of the whole organoids. Overall, the coverage area and perimeter of the 0 μm group expanded more quickly than those of the 0.65, 0.7, and 0.75 μm groups after Day 30 (**Figs 1B and S1A**). A massive increase in human cerebral neocortex size is accompanied by cortical area expansion and the emergence of extensive cortical folds and has been observed in cortical organoids [18–20]. The greater volume may imply that Pur patterning differs between dorsal and ventral fates. To further distinguish dorsal and ventral patterns, we detected the forebrain progenitor marker PAX6 and the cortical neuron marker SATB2 in the sections of organoids with 0 and 0.65 μm Pur at 45 days (**Fig 1E and 1F**) and found that PAX6+ cells were located in the rosettes of the group without Pur, while no PAX6+ cells were seen in the rosettes of the 0.65 μm Pur group (**Fig 1E**). The count of PAX6

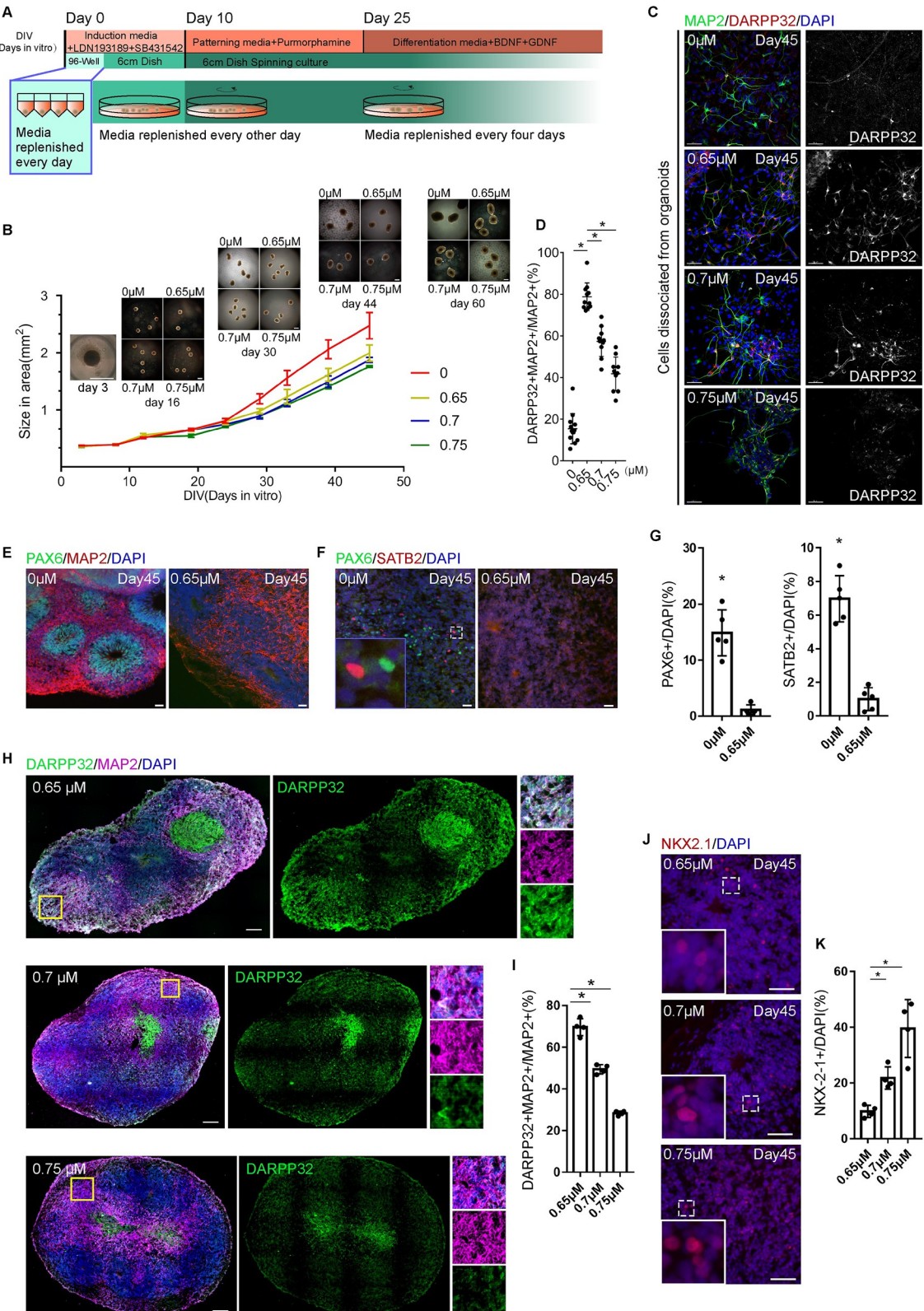

**Fig 1. Generating human striatal organoids. (A)** The schematic of brain organoid protocol with a ventral drug-patterning application (purmorphamine) during culture. **(B)** The morphology of organoids with different concentrations of ventral drug-patterning applications in culture. Scale bar, 1 mm. Quantification of organoid size from Day 0 to Day 44 in cultured (mean ± SD,

$n$ = 4 organoids). **(C, D)** Immunostaining and quantifying DARPP32 and MAP2 antibodies revealed the effects of Pur in striatal-fate patterning in dissociated neurons from organoids ($n$ = 6 organoids). Data, mean ± SD. One-way ANOVA. *, $P < 0.05$. Scale bar, 50 μm. **(E–G)** Immunostaining for PAX6, SATB2, and MAP2 antibodies revealed dorsal and ventral fates distinguished by Pur addition to organoids after Day 45 in culture. Scale bar, 50 μm. Quantifying the expression levels of PAX6 and SATB2 in sections after 45 days in culture. Organoids, $n$ = 5. Data, mean ± SD. Student $t$ test. *, $P < 0.05$. **(H, I)** Immunostaining and quantification for DARPP32 and MAP2 antibodies revealed that the decreased striatal fates with increasing Pur in Day 45 organoids ($n$ = 4 organoids). Data, mean ± SD. One-way ANOVA. *, $P < 0.05$. Scale bar, 50 μm. **(J, K)** Immunostaining and quantification for NKX2.1 antibody revealed that the patterned fates shifted from LGE to MGE, increasing Pur on Day 45 ($n$ = 4 organoids). Data, mean ± SD. One-way ANOVA. *, $P < 0.05$. Scale bar, 50 μm. The raw data underlying this figure can be found in the S1 Data. LGE, lateral ganglionic eminence; MGE, medial ganglionic eminence.

+ cells in the 0 μm group (14.69 ± 4.856%) was higher than that in the 0.65 μm group (0.7214 ± 1.398%), while the count of SATB2+ cells in the 0 μm group (6.758 ± 0.4711%) was higher than that in the 0.65 μm group (0.4623 ± 0.1569%) (**Fig 1G**). We also used qRT–PCR to analyze the transcription levels of *PAX6*, *SATB2*, and *TBR1* (another cortical layer neuron marker). The data showed that the transcription levels of *PAX6*, *SATB2*, and *TBR1* in the group without Pur were higher than those treated with 0.65 μm Pur (**S1B Fig**). These results demonstrated that Pur treatment distinguished dorsal fates from ventral fates. Next, we tested how Pur dosages affected LGE patterning. We detected the expression of the MSN marker DARPP32 and the medial ganglionic eminence (MGE) marker NKX2.1 in the sections of organoids treated with 0.65, 0.7, and 0.75 μm Pur (**Fig 1H and 1I**). At 45 days, DARPP32+ and MAP2+ neurons in the 0.65 μm Pur group were present at 70.67 ± 1.203% abundance, which was significantly higher than the abundance levels of 49.35 ± 0.9497% and 28.2 ± 0.9629% in the 0.7 and 0.75 μm Pur groups, respectively (**Fig 1I**). Instead, NKX2.1+ cells accounted for 40.66 ± 2.085% in the 0.75 μm group, which was significantly higher than the 21.36 ± 1.008% and 9.78 ± 0.5363% abundance levels in the 0.7 and 0.65 μm groups, respectively (**Fig 1J and 1K**). The increase in Pur dosage increased *NKX2.1* transcription, as determined by qRT–PCR (**S1C Fig**). Our results demonstrated that Pur transforms LGE to MGE patterning in a dose-dependent manner in the range of 0.65 to 0.75 μm. Regarding the 2D differentiation method, 0.65 μm Pur could still be better for differentiating hStrOs in strict adherence to the differentiation in our protocol.

## hStrO development mimics LGE development

LGE of the fetal, ventral forebrain becomes striatum [14]. To confirm the forebrain development in hStrOs, we checked the expression levels of neuroectoderm markers PAX6, SOX1, as well as those of forebrain markers OTX2, FOXG1 in cells dissociated from early hStrOs. By Day 20, 88.58 ± 0.05% of the cells dissociated from hStrOs were double-positive for PAX6 and SOX1, and 85.05 ± 0.03% were OTX2 and FOXG1 double-positive (**S2A Fig**). Histological analysis revealed neural rosette-like structures that resemble the proliferative regions of the human VZ in hStrOs (**Figs 2A and S2B**). Each rosette exhibited regular radial organization, including the reduction of SOX2+/Ki67+ progenitors along the apical-basal axis coupled with increasing MAP2+ mature neurons (**Fig 2B**). We further observed the changes in the rosette area during hStrOs development. The thickness of the VZ-like region increased from 47.00 ± 13.04 μm at 30 days to 54.57 ± 17.33 μm at 45 days, while it decreased from 54.57 ± 17.33 μm at 45 days to 41.59 ± 8.499 μm at 80 days (**S2B Fig**). Meanwhile, quantification showed a decrease in SOX2+ neural progenitor cells from Day 30 to Day 80 and an increase in NEUN+ mature neurons from Day 45 to Day 80 (**S2B and S2C Fig**). Although the overall thickness of the VZ-like area in hStrOs was thinner than that in the developing human brain, the neuronal inductions in our hStrOs are effective and sustainable.

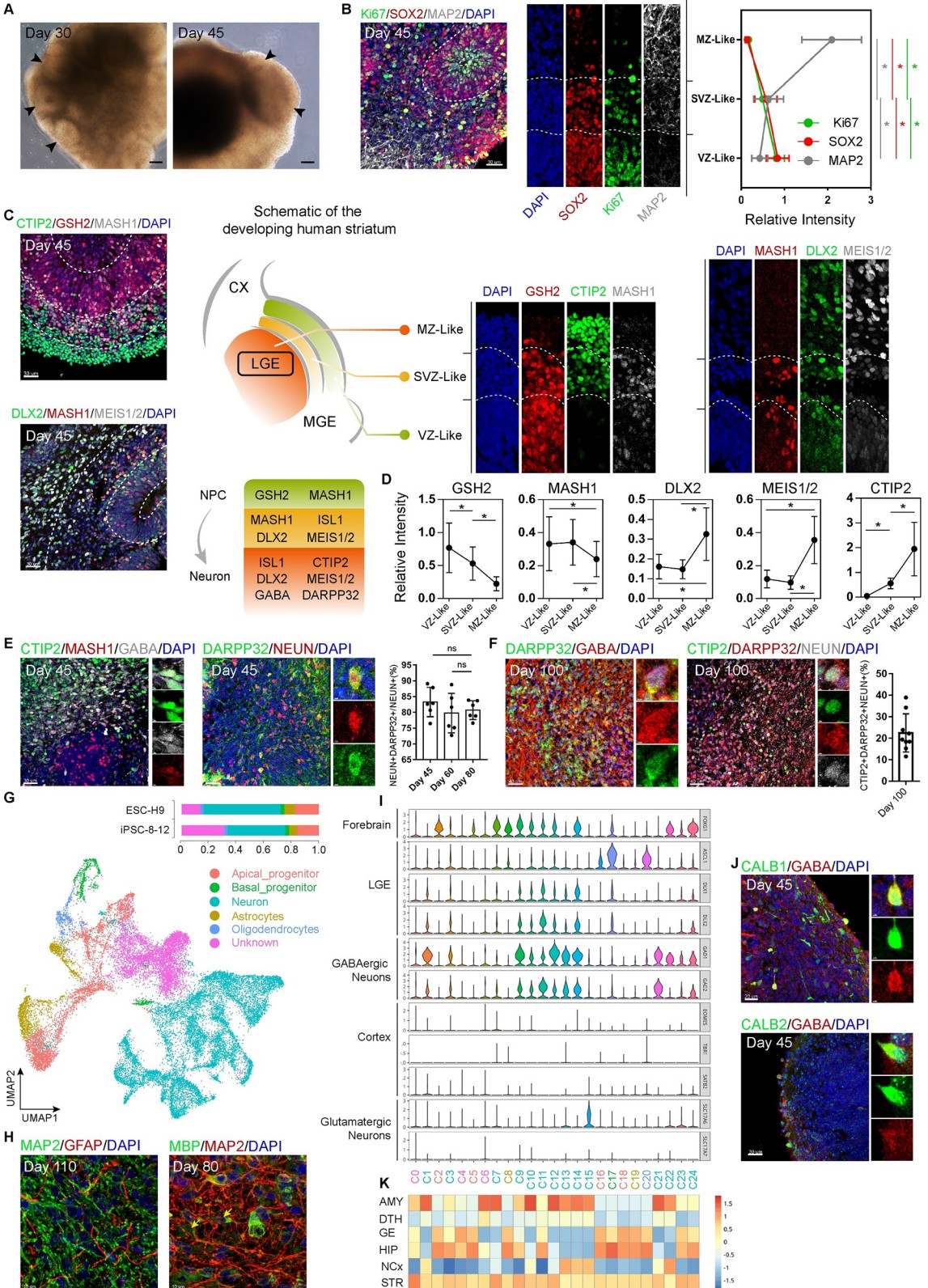

**Fig 2. hStrO development mimics LGE development. (A)** Typical bright-field images of the hStrO were captured on Days 30 and 45 (arrows, the neural rosette-like structures). Scale bar, 100 μm. **(B)** Immunostaining for Ki67, SOX2, and MAP2 antibodies in Day 45

hStrOs. Dashed lines marked the VZ/SVZ/MZ-like zones based on marker distribution and internal cellular organization. Quantifying the Ki67, SOX2, and MAP2 fluorescence intensity in supposed VZ/SVZ/MZ-like zones ($n$ = 8 organoids). Scale bar, 30 μm. Data, mean ± SD. One-way ANOVA. *, $P < 0.05$. **(C)** Immunostaining for GSH2, DLX2, MEIS1/2, MASH1, and CTIP2 antibodies revealed the typical interior cellular organization of Day 45 hStrOs. Schematic representation of a coronal hemisection of the developing brain that represents the NCx, LGE, and MGE, indicating the regional expression of the patterning markers. Dashed lines marked the VZ/SVZ/MZ-like zones based on marker distributions and internal cellular organization. Scale bar, 30 μm. **(D)** Quantifications of the GSH2, MASH1, DLX2, MEIS1/2, and CTIP2 fluorescence intensity in supposed VZ/SVZ/MZ-like zones, respectively ($n$ = 8–10 organoids). Data, mean ± SD. One-way ANOVA. *, $P < 0.05$. **(E)** Immunostainings for MASH1, CTIP2, and GABA antibodies revealed the developing MSNs in Day 45 hStrOs (left). Inserts showed typical MASH1+CTIP2+GABA+ cells. Scale bar, 30 μm. Immunostaining for NEUN and DARPP32 antibodies revealed mature striatal MSNs in Day 45 hStrOs (right). Insert showed typical DARPP32+NEUN+ mature striatal MSNs. Scale bar, 30 μm. Quantification of the DARPP32+NEUN+ cells in NEUN+ cells ($n$ = 6 organoids). Data, mean ± SD. One-way ANOVA. ns, none significant. **(F)** Immunostaining for DARPP32, GABA, CTIP2, and NEUN antibodies revealed mature striatal MSNs in Day 100 hStrOs. Insert showed typical DARPP32+GABA+ cells (left) and CTIP2+DARPP32+NEUN+ cells (right). Scale bar, 40 μm. Quantification of the CTIP2+DARPP32+NEUN+ cells in Day 100 hStrO ($n$ = 9 organoids). The results are normalized by NEUN+ cells: 22.44 ± 8.848%. Data, mean ± SD. **(G)** UMAP visualization of single-cell RNA expression in hStrOs at Day 110 in vitro differentiation ($n$ = 25,963 cells from hESC-H9 and hiPSCs-8-12). UMAP visualization showed 5 major cell types. Histogram showing the percentage of cells in each of the 2 cell lines belonging to each cluster in hStrOs. **(H)** Immunostaining for GFAP, MBP, and MAP2 antibodies revealed astrocytes (Day 110) and oligodendrocytes (Day 80) in hStrOs. Scale bar, 10 μm. **(I)** Expression patterns of key transcription factors among 24 clusters. **(J)** Immunostaining for CALB1, CALB2, and GABA antibodies in Day 45 hStrOs. Insert showed typical CALB1+GABA+ or CALB2+GABA+ cells. Scale bar, 30 μm; insert, 2 μm. **(K)** The correlation with the BrainSpan dataset of the developing human brain (PCW 8, 9, 12, 13, and 16) using the VoxHunt algorithm. The raw data underlying this figure can be found in the S1 Data. AMY, amygdala; DTH, dorsal thalamus; GE ganglionic eminences; HIP, hippocampus; hStrO, human striatal organoid; LGE, lateral ganglionic eminence; MGE, medial ganglionic eminence; MSN, medium spiny neuron; MZ, mantle zone; NCX, neocortex; STR, striatum; SVZ, subventricular zone; UMAP, uniform manifold approximation and projection; VZ, ventricular zone.

Based on multiple previous works, the following specific markers were used to define the LGE in the fetal forebrain (**S1 Table**). During LGE development in vivo, these markers form progressive, nested patterns that distinguish the ventricular zone (VZ) from those of the subventricular zone (SVZ) and the mantle zone (MZ): GSH2+ progenitor and MASH1+ intermediate progenitor cells enriched in VZ and SVZ and striatal MSN markers, such as DLX2, MEIS1/2, and CTIP2, enriched in MZ (**Fig 2C**) [21]. To determine whether the developing hStrOs recapitulated the patterns in developing LGE, we checked the expression of LGE-specific markers on Day 45 hStrOs. These markers exhibited superior radial patterns in hStrOs: GSH2+ cells were mostly located in the rosettes, which we also termed as VZ-like zones; DLX2, MEIS1/2, and CTIP2, excluded from VZ-like zones, increased their expression radially (**Fig 2C and 2D**). Notably, a MASH1+ population was located outside the visible region of the rosette, which possibly signified an SVZ-like region in hStrOs (**Fig 2C and 2D**).

The robust recapitulation of developing LGE was manifested by the existence of striatal GABAergic MSNs in the hStrOs. We identified typical MASH1+/CTIP2+/GABA+ neurons in Day 45 hStrOs indicating the potential developing MSNs in the hStrO (**Fig 2E**), and DARPP32+ cells accounted for 70.67 ± 9.005% of MAP2+ cells in Day 45 hStrOs (**Fig 1I**). To evaluate striatal MSNs in long-term, cultured hStrOs, we stained DARPP32 and NEUN antibodies in Days 45, 60 and 80 hStrOs and identified stable proportions of DARPP32+ neurons in hStrO: Day 45, 82.2 ± 9.713%; Day 60, 80 ± 8.959%; and Day 80, 80.67 ± 5.62% (**Figs 2E and S2C**). Further characterization revealed the typical MSNs co-expressing DARPP32+/GABA+ or DARPP32+/CTIP2+ on Day 110 hStrOs (**Fig 2F**). Notably, we also observed some DARPP32+ cells that are negative for GABA/CTIP2, implying the possibility of DARPP32+ cells with other neuronal identities might present in hStrO (**Fig 2F**). Finally, we defined the mature MSNs in hStrOs by CTIP2+DARPP32+NEUN+, accounted for 18.27 ± 10.27% of NEUN+ cells on Day 100 (**Fig 2F and S2 Table**).

To further understand cell-type specification in hStrOs, we performed scRNA-seq analysis for hStrOs differentiated for 110 days ($n$ = 25,963 cells from hESC-H9 and hiPSCs-8-12). To characterize each cell, uniform manifold approximation and projection (UMAP)

dimensionally reduced the cluster cells from organoids into 24 clusters according to their transcriptome features (S2D Fig). Three clusters (C0, C4, and C6) were excluded from the analysis and assigned to unknown because of low gene reads (22.78%) (S2E Fig). Expression patterns of unique markers then annotated 5 major cell clusters, including apical progenitors enriched in the early neurogenesis genes *NES*, *HES1* (16.60%), basal progenitors enriched in the intermediate progenitor markers *ASCL1*, *HES6* (3.01%), a major group of neurons highly expressing the general neuronal markers *MAP2*, *STMN2* and *DCX* (48.70%), astrocytes expressing *AQP4* and *SLC1A3* (6.72%), and a small group of oligodendrocytes expressing *OLIG2* (2.17%) (Figs 2G, S2F, and S2G). Histological analysis further verified the gliogenic fate in hStrOs. We immunostained hStrOs at Day 110 with GFAP and MAP2 antibodies and found that MAP2 + neurons were extensively intermingled with GFAP+ astrocytes (Fig 2H). Staining of oligodendrocytes with the marker myelin essential protein (MBP) showed that MBP colocalized with the axons of MAP2+ neurons (Fig 2H). To understand lineage commitment in hStrO, we analyzed the gene expression in hStrOs. The forebrain marker *FOXG1* and LGE markers *ASCL1*, *DLX1*, *and DLX2* were widely expressed, whereas few cells in hStrOs expressed other brain region-specific markers, such as *PAX6*, *EOMES*, *TBR1*, and *SATB2* in the cortex; *LHX6* in MGE; *TCF7L2* and *GBX2* in the thalamus (Figs 2I and S2H). Further analysis revealed the specification of the neural types in hStrOs. Most cells expressed the GABA-synthesizing enzyme genes *GAD1* and *GAD2*, especially in the neuron cluster (Figs 2I and S2H). Only a few glutamatergic neurons expressed *SLC17A6* and *SLC17A7* encoded glutamate transporters (Fig 2I and S2H). Our results suggested that hStrOs mainly include LGE-related cells. We next analyzed the neuronal cluster in hStrOs and found neurons expressing *CALB1* and *CALB2* (S2I Fig). In vivo, CALB1 and CALB2 are 2 further genes expressed in the striatal GABAergic MSNs [22]. Immunostaining in hStrOs also confirmed the expression of CALB1 and CALB2 in GABA+ neurons implying the potential striatal, cellular diversity in hStrOs (Fig 2J). In addition, we mapped scRNA-seq data from hStrOs into the BrainSpan human transcriptomic dataset using VoxHunt [23]. We found that the neuronal clusters in hStrOs significantly correlated with the striatum (STR), and the progenitor showed a high-scaled correlation with ganglionic eminences (GE) when compared to pooled brain samples at postconceptional weeks (PCWs) 8, 9, 12, 13, and 16 (Fig 2K). We further compared the correlations between human brain datasets and hStrOs on different days (Day 60, RNA-seq; Day 80 RNA-seq; and Day 110 scRNA-seq). hStrOs also showed a strong correlation with GE (S2J Fig). The correlation values between hStrO and STR increased when hStrOs were maintained for more days (S2J Fig). Consistent with another approach to striatal organoids, our hStrOs correlated with a developing amygdala (AMY), which may indicate the LGE origin for some neurons in the AMY (Figs 2K and S2J) [16].

## Self-organized regionalization in developing hStrO forming potential compartments

After culturing for 45 days, we observed that most of the hStrOs contained a hyaline patch, which did not appear in the Day 45 hCOs nor in the Day 25 hStrOs (Figs 3A and S3A). We hypothesized that the hyaline patches might be due to the unique, organizational schemes in developing hStrOs. To further see the organizational schemes, we used Ki67 and MAP2 antibodies to detect progenitors and neurons in Day 45 hStrOs (Fig 3B–3D). We first identified a united region with dense nuclei containing 1 or more rosettes, which differed from the discrete regions of low nuclei density (Fig 3B–3D). Ki67+ cells were present in the dense region, both luminally within rosettes and interspersed out of rosettes, but were rarely detected beyond the dense region (Figs 3C, 3D and S3B). The dense region was strictly colocalized with SOX2

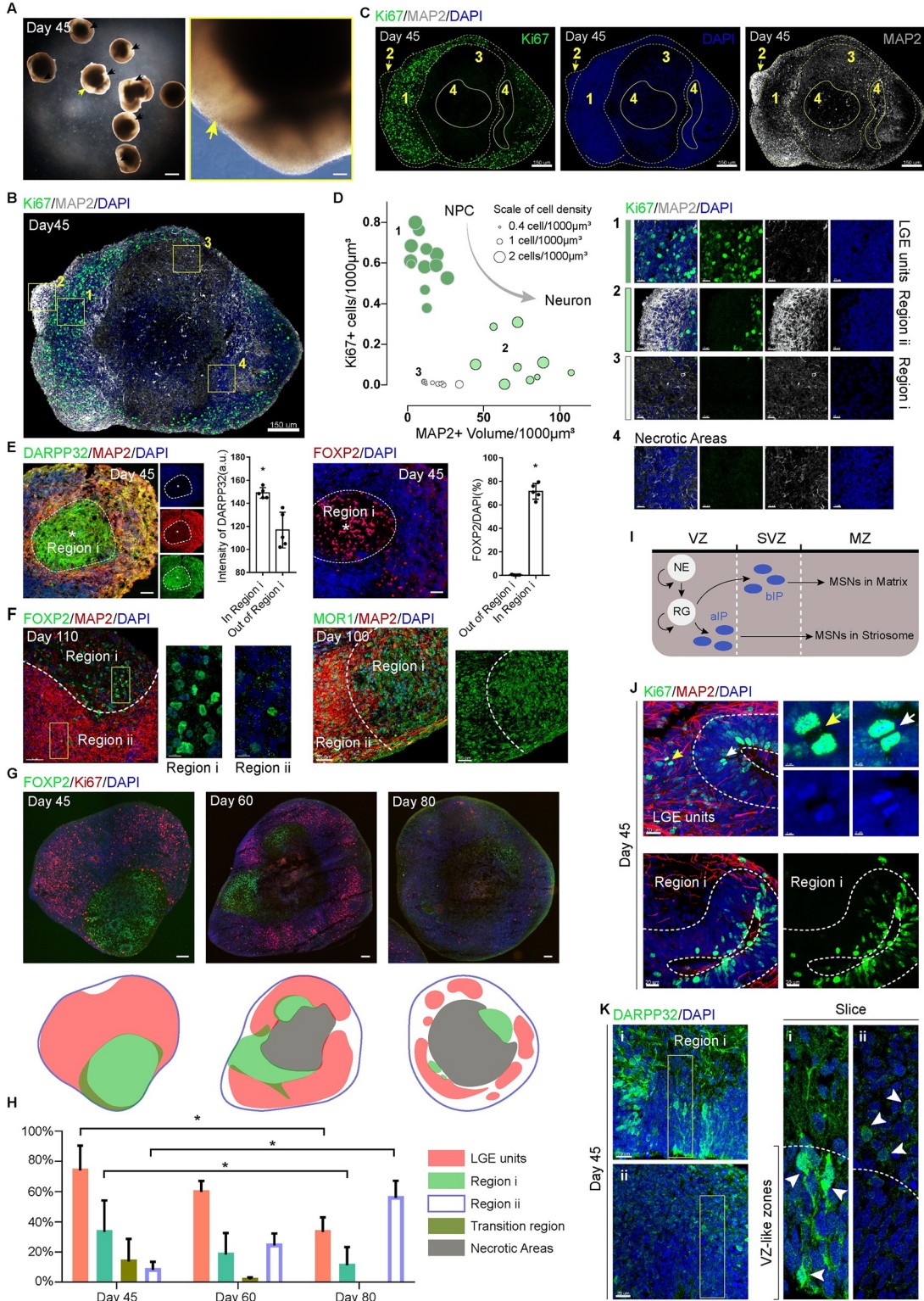

**Fig 3. Self-organized regionalization in developing hStrO forming potential compartments. (A)** Typical bright-field images of the Day 45 hStrOs (arrows, hyaline patch). Scale bar, 1 mm; insert, 100 μm. **(B, C)** Immunostaining for MAP2 and Ki67 revealed a self-organized regionalization in Day 45 hStrOs. Dashed lines marked the supposed region: (1) **LGE units**; (2) **Region ii**; (3) **Region i**; (4) **Necrotic areas**. Scale bar, 150 μm. **(D)** X and Y axis showed the proportion of Ki67+ cells and the volume of MAP2+ dendrites in the supposed regions. Data are from 12 sections of 5 organoids. Some sections do not contain the typical

supposed Region i/ii. The scatters' diameter is adjusted to the target region's cell density. Insert showed typical regionalization as well as necrotic area in Day 45 hStrOs. Scale bar, 15 μm. **(E)** Immunostainings for DARPP32, MAP2, and FOXP2 antibodies revealed the regionalization in Day 45 hStrO distinguished the markers of striatal compartments. The DARPP32/FOXP2 enriched Region i in hStrOs are circled with dashed lines. Scale bar, 50 μm. Quantification showed high DARPP32 and FOXP2 expression in Region i (organoids, $n = 5$). Data, mean ± SD. Student $t$ test. *, $P < 0.05$. **(F)** Immunostainings for MOR1, MAP2, and FOXP2 antibodies revealed the regionalization distinguished by the markers of striatal compartments in Days 100–110 hStrO. Scale bar, 50 μm; insert, 10 μm. **(G)** Immunostainings for Ki67 and FOXP2 antibodies revealed the dynamics of regionalization in hStrO. Scale bar, 50 μm. **(H)** Quantification of the 4 regions divided by Ki67 and FOXP2 ($n = 3$ organoids). Data, mean ± SD. Two-way ANOVA. *, $P < 0.05$. **(I)** A schematic of the topographic organization of striosome and matrix compartments in the development striatum. **(J)** Immunostainings for Ki67 and MAP2 antibodies in LGE units and Region i of Day 45 hStrOs (white arrows, Ki67+ cells in the rosettes; yellow arrows, Ki67+ cells outside the rosettes). Scale bar, 20 μm; insert, 2 μm. **(K)** Immunostainings for DARPP32 antibody in LGE units and Region i of Day 45 hStrOs. Arrows showed that DARPP32 + MSNs are eminent in the rosette close to Region (i) and LGE units (ii). Scale bar, 20 μm. The raw data underlying this figure can be found in the S1 Data. hStrO, human striatal organoid; IP, intermediate progenitor cell; LGE, lateral ganglionic eminence; MSN, medium spiny neuron; MZ, mantle zone; NE, neuroepithelial cell; RG, radioglial cell; SVZ, subventricular zone; VZ, ventricular zone.

expression, and MAP2+ neurons were excluded from the dense region (**Figs 3C, 3D, and S3B**). United regions of dense nuclei were observed from Day 30 to Day 60 in hStrOs (**S3C and S3D Fig**), and the dense region also enriched the striatal, progenitor marker MASH1 and eliminated the striatal, neuron marker GABA (**S4A and S4B Fig**). Our results suggested that the dense-nuclei region, which we termed the LGE unit, represented distinctive in vitro counterparts to the VZ and potential SVZ germinal zones. With the exclusion of necrotic regions in hStrOs, areas beyond the LGE unit can be distinguished into 2 parts based on cellular density and volume of MAP2+ fibers, more likely mirrored MZ regions: Region i has the lowest cellular density and sparse MAP2+ fibers forming hyaline patches in the brightfield; and Region ii has higher cellular density as well as dense+ MAP2+ fibers (**Fig 3D**). Moreover, both Regions i and ii contained GABA+ neurons (**S4C and S4D Fig**). Our results suggested self-organized regionalization in the developing hStrOs.

The regionalization in the postmitotic areas of hStrO is reminiscent of striking compartmental organization in the human striatum: Labyrinthine striosomes embedded within the larger matrix compartment contribute to the striatal functional input and output connectivity. Compartments in the striatum are a histochemically defined in organization [24]. In the developing striatum, DARPP32 and FOXP2 were highly expressed in the striosomes [25–27]. We verified expression of FOXP2 and DARPP32 in serial sections of the fetal brain at 22 W and observed that FOXP2 and DARPP32 were enriched in the striosomes, forming conspicuous mosaics embedded within the striatum, even though they were also detected in the matrix (**S5A and S5B Fig**). To observe whether the regionalization in hStrOs can be delineated by the striatal compartment markers, we detected Day 45 hStrOs using DARPP32 or FOXP2 by costaining with MAP2. Region i enriched in DARPP32 or FOXP2 in stitching images (**S5C and S5D Fig**). Serial section staining also showed strong enrichments of FOXP2 and DARPP32 in Region i (**Fig 3E**). Coimmunostaining with FOXP2 and MAP2 antibodies revealed the stable enrichment of FOXP2 in Region i on Day 110 (**Fig 3F**). The common, striosome marker MOR1 is also enriched in Region i of the 100-day hStrO (**Fig 3F**). To determine whether self-organized regionalization prevails in hStrOs, we sampled a random batch of hStrO on Day 40 and found that 89% of organoids (31/35) had a significant hyaline patch, and 79% of organoids (19/24) showed a clear FOXP2-enriched Region i in random batches of hStrOs on Day 45 (**S6A–S6D Fig**). Notably, the regionalization in the Pur-treated groups but not in the non-Pur-treated group implies that the sonic hedgehog-pathway activation drove the regionalization in hStrO (**S6E Fig**).

In the developing brain, compartmentalized striatogenesis occurs sequentially, precedes matrix formation, and overlaps partially and spatially with the matrix during formation [28–

30]. To evaluate the regionalization dynamics in hStrOs, we used Ki67 and FOXP2 to characterize the regionalization processes in Day 45, Day 60, and Day 80 hStrOs. Based on the patterns of expressions of Ki67 and FOXP2, hStrO was divided into 4 regions: (1) LGE units: enriched Ki67+ cells without FOXP2+ cells; (2) Region i: enriched FOXP2+ cells without Ki67 + cells; (3) Region ii: speckled FOXP2+ cells without Ki67+ cells; and (4) Transition region: mixed Ki67+ cells and FOXP2+ cells (**Figs 3G and S7A**). The size of LGE units and Region i declined from Day 45 to Day 80 paralleling the expansion of Region ii (**Fig 3H**). Moreover, most regions with no Ki67+ cells were Region i on Day 45, and the phenomena lasted until Day 80 in Region ii (**Fig 3G and 3H**). NEUN+DARPP32+ mature MSNs appeared in 2 regions from Day 45 to Day 80 (**S7B Fig**). Our results indicated that Region i formed earlier than Region ii in developing hStrOs.

Neuronal maturation hallmarks are forming synaptic contacts, acquiring spontaneous firing activity, producing dendritic spines, and transmitting nerve impulses along the network [31]. To distinguish neuronal maturation in the regionalization of hStrOs, we detected the establishment of synaptic connections in 2 mature regions from Day 45 to Day 110. At each differentiation point, we quantified the expression of Bassoon or PSD95 puncta in Regions i and ii by coimmunostaining with Tuj1 (**S7C and S7D Fig**). After adjusting the Bassoon +/PSD95+ puncta count by volumes of Tuj1+ fibers, we found that both presynaptic protein Bassoon and postsynaptic protein PSD95 reached their peak in Region i: Bassoon+ puncta from Day 80 to Day 110 showed no difference, and PSD95+ puncta showed no significance from Day 80 to Day 110 and from Day 60 to Day 110 (**S7D Fig**). In Region ii, the number of Bassoon+ puncta on Day 110 was significantly higher than that on Day 80 ($P < 0.0001$). Although PSD95+ puncta were steady from Day 80 to Day 110 in Region ii, the count on Day 110 was significantly higher than that on Day 60 ($P = 0.006$) (**S7D Fig**). These results indicated that the formation of synaptic contacts in the 2 regions was regulated sequentially suggesting sequential regionalization in hStrOs.

Regulated progression of striatal progenitor lineages creates the striatal compartments [29,32]. In detail, a set of apical intermediate progenitors in the VZ generate striosomal MSNs, and another set of basal, intermediate progenitors in the SVZ are fate restricted to matrix MSNs. LGE units in hStrO have extra areas than rosettes similar to the large SVZ zone of the LGE in vivo (**Fig 3B–3D**). In LGE units, we further identified Ki67+ cells in the rosette inner and outer regions (**Fig 3J**). Remarkably, most Ki67+ cells were restricted to the rosettes near Region i, implying that the diversity of progenitors contributes to hStrO regionalization (**Fig 3J**). Meanwhile, DARPP32 staining revealed postmitotic MSNs in regionalized hStrOs (**Fig 2E**). Contrary to DARPP32+ cells scattered in LGE units, 1 group of DARPP32+ MSNs is prominent in the rosette close to Region i, implying that these cells might migrate into Region i (**Fig 3K**). Our results suggested that hStrO regionalization partially recapitulates striatal, compartmental, and MSN-specific rules. Notably, the progenitor, cellular diversity in hStrOs is not spatially confined by regionalization. We also recognized typical, neural tubes that contribute to Region i and LGE units revealing an equal opportunity for the individual rosette to participate in hStrO regionalization (**S8 Fig**).

## Dynamic regionalization in developing hStrO

CALB1 immunoreactive neuropil were enriched in the matrix zones, whereas CALB2 immunoreactive neuropil were enriched in the striosome zones [33,34]. The expression of CALB1 and CALB2 showed an irregular pattern on Day 45 hStrOs, but CALB1 was enriched in Region ii while CALB2 was enriched in Region i on Day 100 hStrOs (**Figs 4A, S9A and S9B**). These patterns differed from FOXP2 expression that persistently delineates hStrO regionalization

indicating that Region i/ii may represent the initial state of striatal compartmentalization in vivo. To verify the immature compartmentalization in Regions i/ii, we detected SOX2+ progenitor cells and MAP2+ mature neurons in Regions i/ii. SOX2+ progenitor cells decreased both in Regions i and ii from 45 days to 100 days coupled with the increase in MAP2+ fibers, which indicate the persistence of postmitotic cells in Regions i/ii (**Fig 4B and 4C**). We also observed that mature neurons preferred to cluster at the edge of Region i and Region ii in Day 60 hStrOs (**S10A–S10D Fig**). The position of mature neurons revealed a relative displacement of mature neurons within Region i/ii: From Day 45 to Day 100, NEUN+ neurons approximated the boundary of Region i, while NEUN+ neurons in Region ii preferred to stay away from its boundaries gradually stacking from outside to inside (**Fig 4D**). DCX is an immature, neuronal marker [35,36]. To determine how the postmitotic behavior of cells is related to regionalization, we used DCX to characterize the immature neurons in Regions i/ii and observed a migratory-like stream of DCX+ cells either toward or parallel to the surface of hStrOs (**Fig 4E and 4F**). Further characterization revealed that some DCX+ cells express GABA but not MAP2 implying the migration of immature, GABAergic neurons in the hStrO (**Fig 4G**).

In addition, EdU tracing showed that EdU predominantly identified progenitor cells settled in the rosette while all EdU+ cells were excluded from the rosette after 10 days implying that EdU+ cells escaped from progenitor fate during migration (**S11A and S11B Fig**). EdU+ cells were found in both LGE units and Region i/ii and were scattered in Region i/ii, suggesting that the complete migration of progenitor cells occurs during the maturation in hStrO (**S11C Fig**). Collectively, our results support that the Region i/ii in hStrO recapitulated the primary, striatal compartments. Region i/ii comprised migrating, immature neurons and striosome/matrix-like zones. These authentic striosome/matrix-like zones were generated and integrated by migrating neurons (**Fig 4H**).

## Fused organoids reconstructed the projection target striatum in vitro

The striatum is the primary input side of the basal ganglia, is involved in complex neural circuits, and receives afferent projections from multiple brain regions, such as the cerebral cortex and nigral [26]. Fused organoids would offer numerous opportunities to study human striatal, neural circuits. Given the proven programs generating hCOs and human midbrain organoids (hMOs), efforts were undertaken to reconstruct the projection target striatum in vitro [8,37]. We generated hCOs and hMOs using hESC-H9 cells expressing green fluorescent protein (GFP) using published methods. Expressions of the cortical progenitor cell marker PAX6 and the cortical intermediate progenitor cell marker TBR2 demonstrated the cortical fate of the hCOs (**S12A Fig**). We further observed TBR1+, CTIP2+, and SATB2+ cells in hCOs indicating the specification of deep and upper cortical layers (**S12B Fig**). In hMOs, we observed FOXA2 + progenitor zones and TH+ dopaminergic neurons (**S12C Fig**). We further identified clusters of FOXA2+TH+ cells, which indicated midbrain, dopaminergic neurons in hMOs (**S12C Fig**).

To generate fused organoids, we fused GFP+ hCO with GFP- hStrO (hC-StrO) and GFP + hMO with GFP- hStrO (hM-StrO) at Day 27, respectively (**Fig 5A**). For comparison, GFP +hStrO and GFP-hStrO were fused simultaneously (hStr-StrO) (**Fig 5A**). We recorded the fusion processes by epifluorescence microscopy at 20 days post-fusion (dpf). At dpf 15, we observed GFP+ axons extended into GFP- hStrO in hC-StrO and hM-StrO. Similar extended GFP+ axons did not appear in hStr-StrO. Instead, we observed migrated GFP+ cells in GFP-hStrO (**Figs 5B and S12D**). Serial sections of fused organoids at 20 dpf confirmed that GFP + axons of hCOs and hMOs formed projections and targeted the fused hStrO (**Fig 5C–5E**). In contrast, few visual projections are formed in hStr-StrO (**Fig 5C–5E**). We observed migrating

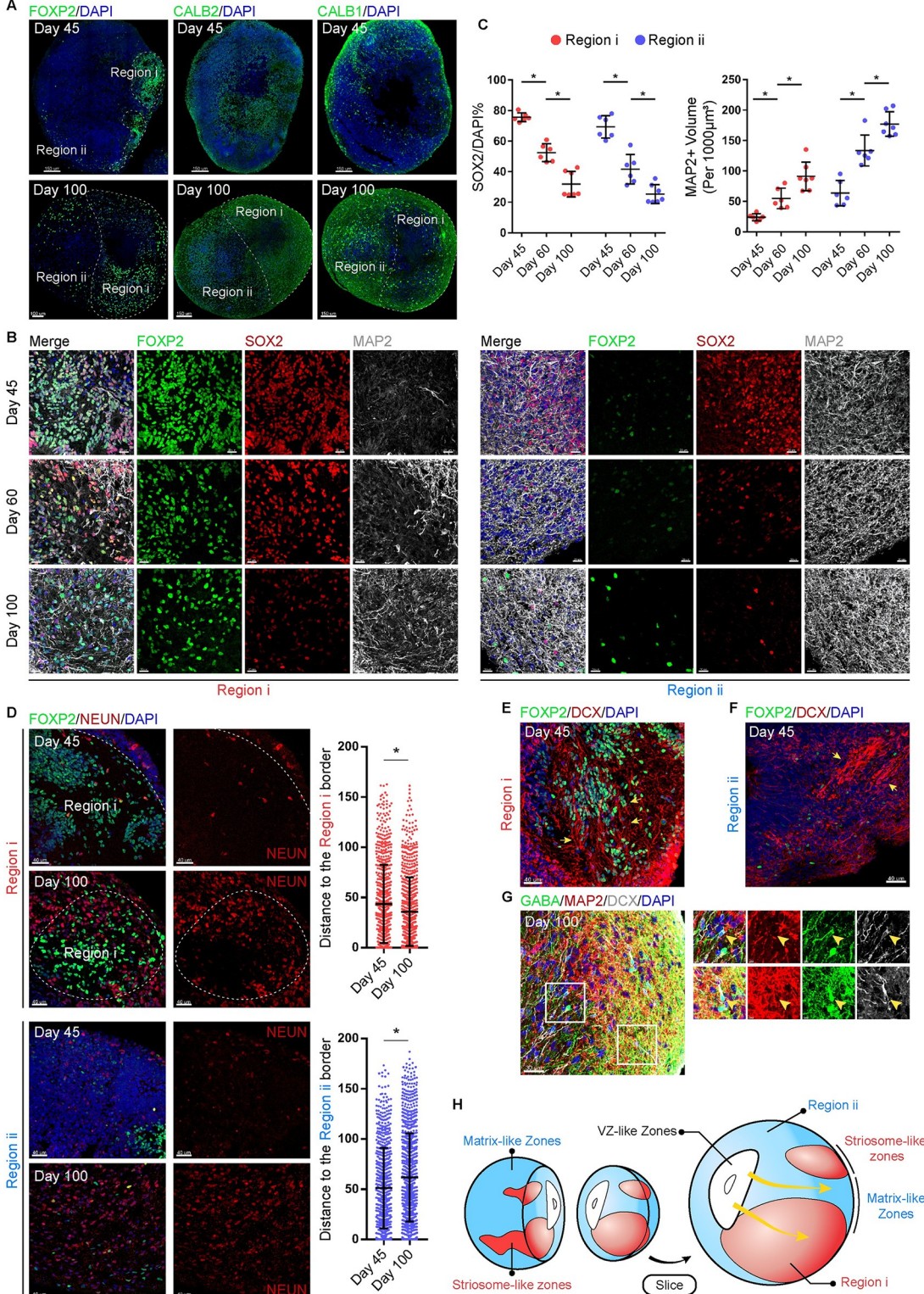

**Fig 4. Dynamic regionalization in developing hStrO. (A)** Immunostainings for CALB1, CALB2, and FOXP2 antibodies in Day 45 and Day 100 hStrOs. Scale bar, 150 μm. **(B, C)** Immunostainings for SOX2, MAP2, and FOXP2 in Day 45, Day 60, and Day 100 hStrOs. Images were collected from Regions i and ii, respectively. Quantifying the proportion of SOX2+ cells and the volume of MAP2+ dendrites in the supposed regions. Organoids, $n = 6$–7. Two-way ANOVA. $^*$, $P < 0.05$. Scale bar, 20 μm. **(D)** Immunostainings for NEUN and FOXP2 antibodies in Day 45 and Day 100 hStrOs. Scale bar, 40 μm. Quantification of the

NEUN+ mature neurons location ($n > 700$ neurons). The location of NEUN+ mature neurons is represented as the distance between NEUN+ cells and the Region i/ii border. The measurement was done in IMARIS. Student $t$ test. $^*$, $P < 0.05$. **(E, F)** On Day 45 hStrOs, Immunostainings for FOXP2 and DCX antibodies revealed the DCX+ cells migrating both in Regions i and ii. Arrows showed the migratory-like stream of DCX+ cells. Scale bar, 40 μm. **(G)** Immunostainings for GABA, MAP2, and DCX antibodies revealed immature GABA+DCX+ neurons without MAP2 in Day 100 hStrOs. Scale bar, 30 μm; insert, 5 μm. **(H)** A schematic of the dynamic regionalization in developing hStrO. The raw data underlying this figure can be found in the S1 Data. hStrO, human striatal organoid; VZ, ventricular zone.

GFP+ cells in all 3 groups of fused organoids. We further quantified the areas of GFP+ axons and the counts of GFP+ cells in GFP- hStrOs. We found that, compared with the area of GFP + axons in hStr-StrOs, the area of GFP+ axons in both hC-StrOs and hM-StrOs increased (**Fig 5F**). Meanwhile, there was no significant in the number of GFP+ cells settled in GFP- hStrO among all 3 groups (**Fig 5F**). The results indicated that distinguished from hStr-StrO, hC-StrO and hM-StrO reconstructed the projection target striatum in vitro.

Remarkably, we observed that the axons from hCO had a different preference for pathfinding than axons from hMO (**Fig 5E**). Referring to the previously described methods [11], axonal distribution was quantified as the percentage of axons protruding into brain organoids based on their extension into the proximal site (R1), if they reached the middle (R2), or if they reached the distal end (R3) of the brain organoid. GFP+ axons were quantified at dpf 20 in hC-StrOs and hM-StrOs in which axons were significantly enriched in R1 of hC-StrOs and uniformly distributed in hM-StrOs (**Fig 5G–5I**). Our results indicated unique projections targeting hStrOs from other brain region-specific organoids in vitro and suggested that the axonal pathfinding appears nonrandom. Environmental cues guide their routes because the GFP + projection-enriched region in hC-StrOs overlaps with the FOXP2+ Region i derived from regionalization in hStrOs (**Fig 5J**). Based on the high similitude between self-organized regionalization of hStrOs and the compartmentalization in the human striatum, it is rational to consider these results as reminiscent of early cortical innervation appearing to match the striosome location in vivo [26]. We also detected hC-StrOs and hM-StrOs by coimmunostaining GABA with GFP antibodies. GFP+ axons extensively intermingled with GABA+ neurons in hStrOs, implying the formation of synaptic connections between them (**S12E and S12F Fig**). To assess the synaptic connections, we immunostained the fused organoid with Bassoon and PSD95 antibody at dpf 20 and observed the conjugation of pre- and postsynaptic puncta in hC-StrOs and hC-StrOs (**S12G and S12H Fig**). We also noticed that the GFP+ fibers from hCOs or hMOs protruding into the hC-StrOs were proximate to the PSD95 puncta on GABA +GFP- fibers from hStrO (**S12I and S12J Fig**). These results suggested that projections targeting hStrO may form synapses in the fused organoids. Together, our data demonstrate that fused organoids serve as a model for projection targeting the striatum in vitro.

## Applying the hStrO model to Huntington's disease

Huntington's disease (HD) is an inherited and late-onset neurodegenerative disorder caused by a CAG-repeat expansion within the Huntingtin (*HTT*) [38]. Early developmental deficits and alterations in brain structures, including to the striatum, were observed in the fetal HD brain [15,39,40]. To determine whether the developmental deficits also appeared in HD hStrOs, we constructed the hStrOs of 3 hiPSCs (CAG, 59; CAG, 55; CAG, 19) derived from a large HD family and observed that the size of hStrO in the HD group was smaller than that in both healthy siblings and hESC-H9 beginning on Day 25 (**Fig 6A and 6B**). SOX2 staining showed that rosettes and neural progenitors in the HD group were smaller and fewer than those in the healthy siblings and hESC-H9 cells from Day 30 to Day 47 (**Fig 6C and 6D**). To characterize the ventralizing efficiency in HD hStrOs, we colabeled hStrOs with GSH2,

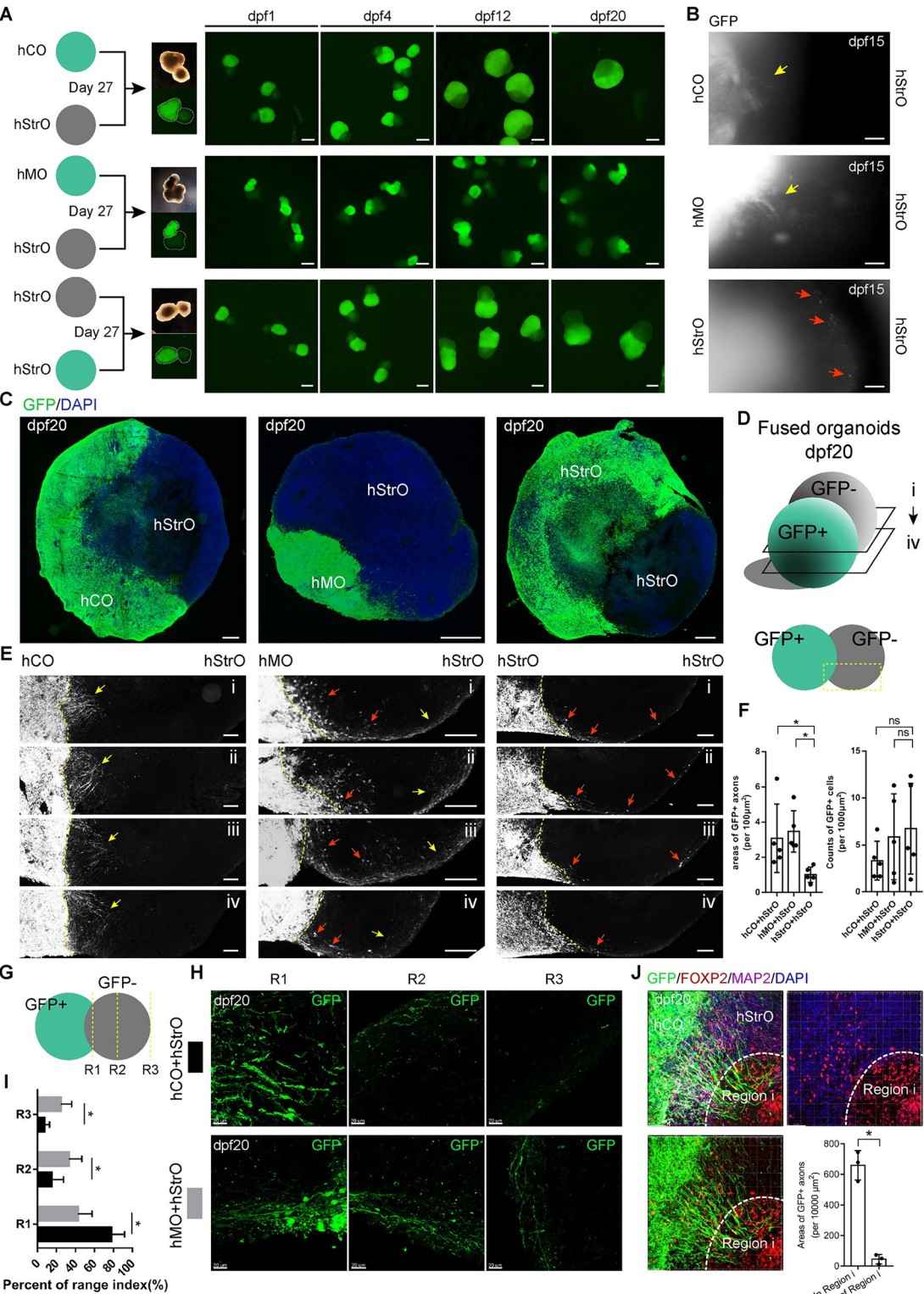

**Fig 5. Fused organoids reconstructed the projection target striatum in vitro. (A)** Schematic view showing the generation of fused organoids. Fluorescent images of fused organoids within dpf 20. Scale bar, 1 mm. **(B)** Higher magnification of fused organoids during culture. Yellow arrows showed GFP+ axons projected out from hCO and hMO; red arrows showed GFP + migration cells from hStrO. Scale bar, 100 μm. **(C)** Stitching images of fused organoids at dpf 20. Scale bar, 200 μm. **(D)** Schematic view of serial sections observed in fused organoids. **(E)** Serial sections of fused organoids revealed GFP+ axons

targeting specific areas of fused hC-StrOs and hM-StrOs, which cannot be noticed in fused hStr-StrOs (yellow arrows, GFP + axons; red arrows, GFP+ migration cells) Scale bar, 100 μm. **(F)** Quantification of the GFP+ axons and the GFP+ cells in fused organoids at dpf 20 (*n* = 5 fused organoids). Data, mean ± SD. One-way ANOVA. *, *P* < 0.05; ns, none significant. **(G)** Schematic of the method to count axons. **(H)** Typical images of hC-StrO and hM-StrO in R1, R2, and R3. Scale bar, 20 μm. **(I)** Quantifying the GFP+ axons in R1, R2, and R3 of dpf 20 fused organoids (fused organoids, *n* = 5). Data, mean ± SD. One-way ANOVA. *, *P* < 0.05. **(J)** Immunostaining for FOXP2, GFP, and MAP2 antibodies in dpf 20 hC-StrOs. The mature Region i in hStrOs are circled with dashed lines in GFP- hStrOs. Quantifying the GFP+ axons in and out of mature Region i (fused organoids, *n* = 3). Data, mean ± SD. Student *t* test. *, *P* < 0.05. Scale bar, 50 μm. The raw data underlying this figure can be found in the S1 Data. hCO, human cortical organoid; hMO, human midbrain organoid; hStrO, human striatal organoid.

MASH1, and CTIP2 antibodies and found that GSH2+ striatal, progenitor cells decreased and MASH1+ striatal, intermediate progenitor cells in HD hStrOs increased compared to healthy hStrOs (**Fig 6E and 6F**). However, the decrease in CTIP2+ cells in HD-hStrO was insignificant (**Fig 6E and 6F**). CTIP2+ is a marker of striatal, MSN neurons. Thus, we further compared striatal MSNs in the healthy group with the HD group in long-term culture. The ratio of CTIP2+NEUN+DARPP32+ cells/NEUN+ cells representing the proportion of striatal MSNs in HD hStrOs was similar to that in healthy hStrOs on Day 100 (*P* = 0.3683) (**Figs 2F and 6G**). These data indicate that our method to generate hStrOs might be suitable for studying the developmental deficits of the striatum in HD.

To further expand the utility of hStrOs, we injected striatal neurons from patients with HD, derived from hiPSCs, that harbored 55 CAG repeats and stably expressed, GFP into Day 45 hStrOs. After injection, we stopped the spinning for 24 h to promote neuronal survival and continuously monitored HD striatal neuron trajectories with GFP in hStrOs by epifluorescence microscopy from 24 to 192 h posttransplantation (**Fig 6H and 6I**). The transplanted HD neurons elongated their dendrites and branches, and the longest projection extent ranged from 38.02 ± 4.085 μm (24 h) to 145.8 ± 6.536 μm (192 h) (**Fig 6H and 6I**). Next, we quantified neuronal morphology by Sholl analysis to show the complexity of neurons [41]. The results of Sholl analysis showed that after transplantation, the overall neuron complexity increased (**Fig 6I**). During monitoring, the positions of a few neurons shifted (**S13A Fig**). The neuronal mobilizing trajectory map showed that 31.8% of traced neuron (*n* = 22) trajectories were significantly motile (**S13A Fig**). To determine the taxis of migratory neurons, we extended the transplanted neurons and observation time. After expanding the transplanted neurons, we observed significant, neuronal migration in the early days after transplantation (**Fig 6J**). We monitored the transplanted hStrOs for more than 21 days. From 3 days to 21 days after transplantation, we observed that migrating neurons clustered within a particular area reminiscent of the adhering graft in hStrOs regionalization (**Fig 6J**).

We sectioned organoids and stained FOXP2 30 days after transplantation to determine the clustered area of migrating neurons. Many GFP+ cells were found in FOXP2+ mature regions in hStrOs, which presented as a distinct, migrating streak located in the interior of Region i (**Fig 6K–6M**). GFP+ cells also settled on the surface of host hStrOs (**Fig 6K–6M**). Our results implied that transplanted neurons might have the same developmental processes in the host hStrOs. Thirty days after transplantation, to observe the potential integration of transplanted neurons, we stained presynaptic protein Bassoon and postsynaptic protein PSD95 with Tuj1 (**S13B Fig**). 3D rendering by IMARIS showed entangled GFP+Tuj1+ transplanted neurons with GFP-Tuj1+ host neurons and Bassoon/PSD95 localized on the nerve fibers revealing potential, synaptic connections between transplanted neurons and host neurons (**S13B Fig**). Our results demonstrated that the activity of target neurons could be visualized by reporter genes in the chimeric hStrOs, including growth, migration, and potential integration. In this chimeric hStrO, the target neuron and the host can be flexible. Our model allows us to approach aberrant behavior in individual neurons or colonies that carry HD mutations.

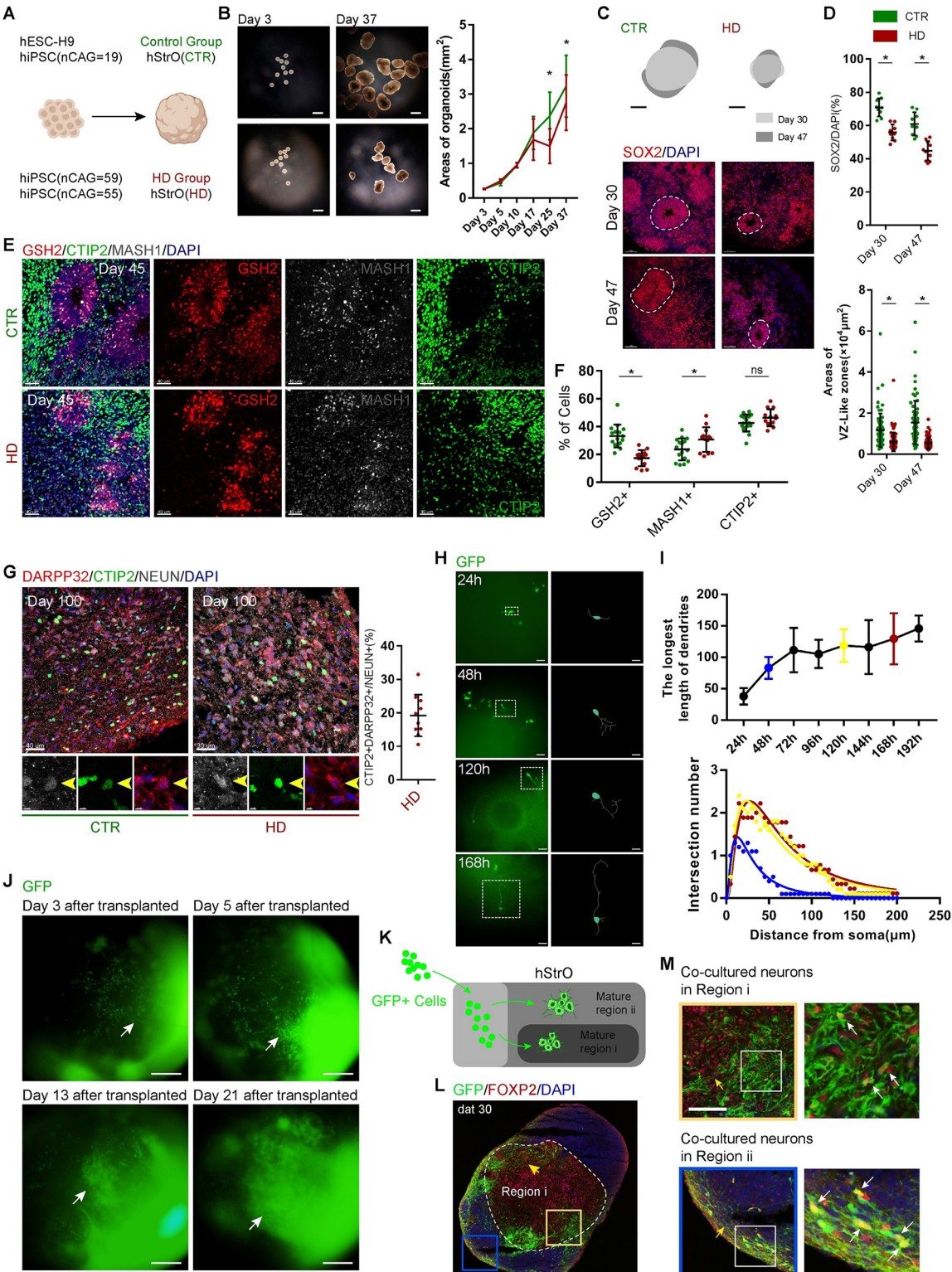

**Fig 6. hStrOs model the striatal development of Huntington's disease. (A)** Schematic of generating CTR hStrOs and HD hStrOs. **(B)** Morphology of CTR and HD hStrOs after 3 days and 37 days of culture. Scale bar, 1 mm. In culture, the quantity of organoid size in the area from Day 3 to Day 37. Data, mean ± SD. Organoids, $n > 10$. One-way ANOVA. *, $P < 0.05$. **(C, D)** Immunostaining for SOX2 antibody on Day 45 CTR and HD hStrOs. Dashed lines marked the rosette in the hStrO. Quantifying SOX2+ cells and rosette (also termed VZ-like zones) areas in hStrOs. Organoids, $n = 9$–10. One-way ANOVA. *, $P < 0.05$. Scale bar, 50 μm. **(E, F)** Immunostaining

and quantification of GSH2, MASH1, and CTIP2 antibodies in Day 45 CTR and HD hStrOs. Organoids, $n$ = 13–16. One-way ANOVA. *, $P$ < 0.05. Scale bar, 50 μm. **(G)** Immunostaining for DARPP32, CTIP2, and NEUN antibodies in Day 100 CTR and HD hStrOs. Quantification of DARPP32+CTIP2+NEUN+ cells in HD hStrOs. Scale bar, left, 40 μm; right, 30 μm. **(H)** Fluorescent images of hStrOs after transplantation. Tracing target neurons revealed their evolving morphology. Scale bar, 100 μm. **(I)** Quantifying the longest synaptic length of a single neuron and the mean synaptic length after coculture ($n$ = 8 neurons from 4 organoids). Data, mean ± SD. Sholl analysis shows that cocultured neurons gradually exhibit more complex morphology after 48 and 120 h. At least 8 neurons from 4 organoids per group were analyzed by Sholl. **(J)** Typical images of hStrOs revealed that GFP+ neurons exhibit migratory-like morphology after coculture. Scale bar, 200 μm. **(K)** Schematic view of the migration of transplanted neurons in host organoids. **(L, M)** Immunostainings for GFP and FOXP2 antibodies revealed the position of GFP+ transplanted neurons in the mature Region i of host hStrOs (yellow arrows, typical migrating streaks; white arrows, GFP+FOXP2+ cells.) Dat, days after transplanted. Scale bar, 100 μm. The raw data underlying this figure can be found in the S1 Data. GFP, green fluorescent protein; HD, Huntington's disease; hStrO, human striatal organoid; VZ, ventricular zone.

## Discussion

Despite the wide reporting of region-specific brain organoids, including Activin-A-patterned human striatal organoids, the protocol to create striatal organoids focusing on striatal cytoarchitecture is still rare. [9–11,16,37]. Here, we describe a method to generate hStrOs based on the classical SFEBq culture [42]. The protocol is easy to follow and depends on a similar, underlying principle to generate 2D striatal medium spiny GABA neurons from human ESCs, which uses early, simultaneous activation of SHH signaling using Pur (0.65 μm) [4].

The percentage of DARPP32+ neurons exceeded 70% in this protocol. Based on the partially overlapping of DARPP32 with GABA/CTIP2, any one marker is never enough to absolutely identify striatal MSNs. In this study, we defined CTIP2+DARPP32+NEUN+ cells (18.27 ± 10.27%) as mature MSNs in Day 100 hStrOs. The partial segregation of mature striatal MSN markers in our hStrO might result from the fluctuation of neuronal markers in the different time lapses of striatal development. Another possibility is that the absence of neural circuits in our hStrO might cause the segregation of striatal markers. Previous studies have demonstrated that CTIP2 knockout did not compromise the generation of MSNs from hESCs [43]. CTIP2 contributed to axonal extension and pathfinding during striatal development [44,45]. The DARPP32+ neurons without CTIP2 in hStrOs could be isolated MSNs similar to MSNs undergoing apoptosis in vivo [46]. Thus, the extent to which the lack of neural circuits in the hStrO contributes to the segregation of striatal markers is worth addressing in the future. Compared to 2D cell culture, organoids model the spatial cytoarchitecture of organs more closely. Superior, radial organization in the hStrOs' rosette reflects the spatial cytoarchitecture of the striatum. The supposed VZ-, SVZ-, and MZ-like zones based on the individual rosette and nested expression of striatal markers that are typical of radial organization existed in hStrO and did not appear in 2D culture. Our study provides a platform for investigating striatal development in vitro in a 3D cellular environment.

Moreover, the closer model is also reflected in the self-organized regionalization of hStrOs. The self-organized regionalization likely derived from the ventralization in hStrO occurred in multiple batches of hStrO and possibly signified compartment-like structures, mimicking the developing striatum. Self-organized regionalization in hStrO forms 2 regions: Region i is similar to the striatal striosome zones and Region ii resembles the matrix zones in vivo. The nature of striatal compartments is to sort the output and input connectivity of the striatal network [47,48]. Thus, the lack of adequate input and output units makes the characterization of compartment-like structures difficult in vitro [26]. However, based on the molecular profiles and developmental process, it is rational to consider the regionalization in hStrOs mimics striatal compartments. Notably, this model is still preliminary. We observed the continuous neurogenesis in the compartment-like structures in hStrOs.

Meanwhile, partial compartment markers, such as CALB1 and CALB2, updated their spatial distribution in hStrO in parallel with neurogenesis. The preliminary compartmentalization

in hStrO might be attributed to 2 reasons: Our hStrOs are still in an early stage-based research on the other organoids [49] and did not reach the stage of advanced striatal compartments, activating the sorter of output and input connectivity in the striatum and the interaction between the striatum and other regions may contribute to the development of striatal compartments as we have seen in fused organoids [15]. Despite the limitation of our hStrO, self-organized regionalization in hStrOs does share similar underlying mechanisms with striatal compartments in vivo. Most researchers used rodents because of the scarcity of models to study human striatal development. The introduction of hStrOs has the potential to overcome interspecies differences. Given the plentiful work of striatal compartments using model animals, in the future, the characterization of neurochemical and electrophysiological properties in hStrOs is required to fully address to what extent the striosome/matrix zones in hStrOs resemble the in vivo structure [29,50]. Referring to the method for characterizing the striatal compartments in rodent models, genetic-based histological and electrophysiological analysis would be helpful to tools [29,50,51].

Using fused organoids, we further recapitulated interregional interactions. Projections from hCOs or hMOs, but not from hStrOs, targeted into hStrOs. Projections from hCOs enriched in the FOXP2+ regions, reminiscent of striosomal MSNs, receive preferential inputs from cortex [26], and projections from hMOs settled equally in hStrO. In the striatum, the striosomal MSNs are innervated by dopaminergic neurons in the ventral tier of the SNc (the A9 cell group) and SNr (dense cellular zone), while dopaminergic neurons preferentially innervate matrix neurons in the dorsal tier of the SNc, the retrorubral area (A8 cell group), and the ventral tegmental area (A10 cell group) [26]. hMOs did not distinguish the dopaminergic neuronal diversity, which may explain the uniform distribution of projections targeting the hStrO in hM-StrOs. Based on our system, further adjustments to the differentiation of organoids in the future might refine the system of fused organoids and drive the application of fused organoids to study the development of striatal circuits.

Furthermore, our system has several potential applications in studying diseases related to human striatal development. hStrO can be used as an in vitro disease model. We have modeled some aspects of HD in hStrOs, including the altered rosette and the proportion of striatal progenitors consistent with what was reported in rodent models [52]. Interestingly, the generation of MSNs is not defective in HD hStrOs. Similarly, studies have shown behavioral recovery following HD-iPSC-derived, neural progenitor cell transplantation into HD animal models [53]. However, the impact of altered neural differentiation in grafts remains unclear. Given the deep location of the striatum, the spatial and temporal activities of neural differentiation in HD are difficult to monitor. To further visualize the HD MSN, we raised a modularity framework, which regarded hStrOs as a sort of window on HD MSN in the basic physiological environment. Intravital monitoring of neuronal activities is an effective way to evaluate cell survival and functional integration. Similar to most chimeric organoids reported, the in vitro transplantation we raised is easy to perform [54,55]. Given the apparent difficulties of studying HD patients' striatum, chimeric hStrOs can be used to predict the HD MSN activities in a dish, which will benefit the assessment of HD patients both in development and after transplantation.

We reported a method to generate human striatal organoids from hPSCs by improved SFEBq culturing. We systemically annotated the consecutive development in our hStrOs system, which provides an enhanced repertoire of phenotypic assays in neurological disorders modeled by hStrOs. To our knowledge, our systems are the first to show self-organized, striatal, compartment-like zones in vitro. By combining the reconstructed projection and the transplanted neurons, our system allows us to explore the orderly neural connectivity, even by

testing the survival of transplanted neurons in the striatum and will open new avenues for potential applications for hStrOs.

## Materials and methods

### Ethics and human ES and iPS maintaining

The ethics were approved (No. 28) by the Ethics Committee of the Institutes of Biomedical Sciences at Fudan University. Human ES cell line H9 and human iPS cell line 8–12 were maintained in feeder-free culture. hiPS cell line 8–12 was derived from the fibroblasts of 19 years old female. Yamanaka's factors (*Oct4*, *Sox2*, *Klf4*, and *c-Myc*) were used to construct hiPS cell line 8–12. All hESCs and hiPSCs were maintained on Matrigel (Corning, 354277) in E8 medium (Gibco, A1517001) and were passaged every 5 to 7 days by EDTA.

### Human tissue

Human fetal tissues were collected from patients that requested pregnancy terminations and autopsy diagnostic procedures, fixed by 4% PFA, and sectioned for immunostaining. All procedures were approved (No. 12) by the Ethics Committee of the Institutes of Biomedical Sciences at Fudan University.

### Human striatal organoids (hStrOs) culture

ES/iPS colonies were dissociated into a single-cell with Accutase (Gibco, A1110501). A total of 9,000 dissociated cells were plated into each well of a V-bottom ultra-low attachment 96-well plate (Sumitomo Bakelite, MS-9096VZ) that contains an induction media with mixture of Human Neural differentiation Medium and E8 media (1:1), 0.3 μm LDN-193189 (STEM-GENT, 040074), 2 μm SB431542 (Ametek Scientific, DM-0970), and 10 μm Y-27632 (APE. BIO, A3008), Human Neural differentiation Medium based on DMEM-F12 (Gibco, C11330500BT), including 1% (v/v) N2 (Gibco, 17502–048), 1% (v/v) MEM-NEAA (Gibco, 11140–050), 1% (v/v) GlutaMAX (Gibco, 35050–061). Cell aggregates had been cultured statically for 3 days in a 37°C incubator with 5% $CO_2$. On Day 3, the organoids were removed to a 60-mm ultra-low attachment plate, and the Y-27632 compound was stopped to supply. Media was replaced every day for 10 days. On Day 10, organoids were transferred to a spinning culture (60 rpm/min). The striatal patterning media contains 1:1 mixture of DMEM-F12 media and Neurobasal media (Gibco, 21103–049) supplemented with 1% (v/v) MEM-NEAA, 1% (v/v) GlutaMAX, 1% (v/v) N2 supplement, 2% (v/v) B27 supplement, 1% (v/v) Penicillin/Streptomycin, and 0.65 μm purmorphamine (APE.BIO, A8228). Neural differentiation media, 1:1 mixture of DMEM-F12 media and Neurobasal media supplemented with 1% (v/v) MEM-NEAA, 1% (v/v) GlutaMAX, 1% (v/v) N2 supplement, 2% (v/v) B27 supplement (Gibco, 117504–044), 1% (v/v) Penicillin/Streptomycin (Gibco, 2051357), 20 ng/ml BDNF, and 10 ng/ml GDNF, was used from Day 25. From Day 45, the BDNF (Peprotech, AF450002) and GDNF (Peprotech, AF450010) were stopped to supply.

### Human cortical organoids (hCOs) and human midbrain organoids (hMOs) culture

hCOs were generated referring previously described method of Pasca et al [8]. To initiate the generation of hCOs, ES/iPS colonies were dissociated into a single-cell with Accutase (Gibco, A1110501). A total of 9,000 dissociated cells were plated into each well of a V-bottom ultra-low attachment 96-well plate (Sumitomo Bakelite, MS-9096VZ). Induction media contained Human Neural differentiation Medium and E8 media (1:1), supplemented with the 2 SMAD

inhibitors 0.3 µm LDN-193189 (STEMGENT, 040074), 2 µm SB431542 (Ametek Scientific, DM-0970), and 10 µm Y-27632 (APE.BIO, A3008). On the sixth day in suspension, hCOs were transferred to neural differentiation media containing 1:1 mixture of DMEM-F12 media and Neurobasal media supplemented with 1% (v/v) MEM-NEAA (Gibco, 11140–050), 1% (v/v) GlutaMAX (Gibco, 35050–061), 1% (v/v) N2 supplement (Gibco, 17502–048), 2% (v/v) B27 supplement (Gibco,), 1% (v/v) Penicillin/Streptomycin (Gibco, 2051357). Growth factors 10 ng/ml EGF (R&D Systems, 236-EG) and 20 ng/ml bFGF (R&D Systems, 233FB/CF) were used from Day 6 to Day 23. Growth factors 20 ng/ml BDNF (Peprotech, AF450002) and 10 ng/ml GDNF (Peprotech, AF450010) were used from Day 24 to Day 45.

hMOs were generated referring previously described method of Qian and colleagues [37]. From Day 1 to Day 4, medium contained Human Neural differentiation Medium and E8 media (1:1), 1% (v/v) GlutaMAX, 100 nM LDN-193189, 10 µm SB431542, 100 ng/ml SHH (R&D Systems, Q62226), 2 µm purmorphamine, and 100 ng/ml FGF8b (Peprotech, 100–25). From Day 5 to Day 6, medium contained DMEM-F12, 1% (v/v) N2, 1% (v/v) GlutaMAX, 100 nM LDN-193189, 10 µm SB431542, 100 ng/ml SHH, 2 µm purmorphamine, 3 µm CHIR99021 (APExBIO, A3011), and 100 ng/ml FGF8b. From Day 7 to Day 13, medium contained DMEM-F12, 1% (v/v) N2, 1% (v/v) GlutaMAX, 100 nM LDN-193189, 100 ng/ml SHH (R&D Systems, Q62226), and 3 µm CHIR99021. From Day 14, medium contained Neurobasal media, 2% (v/v) B27 supplement, 20 ng/ml BDNF, and 20 ng/ml GDNF.

## Dissociation of organoids for 2D culture

Organoids were washed with a DPBS buffer and then incubated with Accutase, shaking at 37˚C water for 4 min. After removing Accutase, the dissociated organoid cells were washed with DMEM-F12 and then plated on a coverslip coated with poly-ornithine (Sigma, 26982-21-8) and laminin (Stemcells, 77003) supplemented the neural differentiation media for culture in an incubator at 37˚C with 5% $CO_2$.

## Generation of a fused organoid

After culturing for 27 days, the 2 organoids were fused by placing them in close proximity in 1.5-ml Eppendorf tubes for 24 h in an incubator. The fused organoids were transferred to 60-mm ultra-low attachment plates in neural differentiation media as previously described.

## Immunostaining

Organoids were fixed by ice cold 4% paraformaldehyde (PFA) for 12 to 16 h. After washing for 3 times with PBS, the fixed organoids were transferred to 30% sucrose for dehydration and then cryosectioned. Coverslip cultures were also fixed in 4% paraformaldehyde for 15 min at room temperature. The immunostaining was done by following our previously described protocol [4]. Primary antibodies used were as follows: anti-Bassoon (mouse, Abcam, AB82958,1:1,000), anti-DARPP32 (rabbit, Chemicon, AB1656, 1:1,000), anti-FOXG1 (rabbit, Abcam, ab18259,1:200), anti-GABA (rabbit, Sigma, A0310, 1:200), anti-GFAP (rabbit, DAKO, Z0334, 1:1,000), anti-DLX2 (rabbit), anti-MEIS1/2 (goat,), anti-Ki67 (mouse, Millipore, LV1825852, 1:500), anti-MAP2 (mouse, Sigma, m1406, 1:1,000; rabbit, Santa Cruz, sc20172, 1:3,000), anti-MASH1 (mouse, BD, 556604,1:500), anti-NKX2.1 (mouse, Chemicon, MAB5460, 1:500), anti-OTX2 (goat, R&D, AF1979, 1:1,000), anti-PAX6 (rabbit, Biolegend, 901301, 1:1,000), anti-SOX2 (mouse, R&D, MAB2018,1:1,000), anti-PSD95 (rabbit, Abcam, ab18258), anti-FOXP2 (rabbit, Abcam, ab16046, 1:2,000), anti-SOX1 (goat, R&D, AF3369, 1:1,000), anti-SATB2 (mouse, Santa Cruz, sc81376,1:200), anti-GSH2 (rabbit, Millipore, 3388451, 1:500), anti-CTIP2 (rat, Abcam, ab18465, 1:1,000), anti-CALB1 (rabbit, Chemicon,

AB1778, 1:500), anti-CALB2 (rabbit, Epitomics, 2624–1, 1:500), anti-DCX (goat, Santa Cruz, SC8066,1:1,000), anti-MOR1 (rabbit, Immunostar, ab572251,1:5,000).

Secondary antibodies used were as follows: Alexa Fluor 488 Donkey anti-mouse IgG (Invitrogen, Molecular Probe, A21202, 1:1,000), Alexa Fluor 594 Donkey anti-mouse IgG (Invitrogen, Molecular Probe, A21203, 1:1,000), Alexa Fluor 594 Donkey anti-rabbit IgG (Invitrogen, Molecular Probe, A21207, 1:1,000), Alexa Fluor 488 Donkey anti-rabbit IgG (Invitrogen, Molecular Probe, A21206, 1:1,000), Alexa Fluor 594 Donkey anti-goat IgG (Invitrogen, Molecular Probe, A11058, 1:1,000), Alexa Fluor 488 Donkey anti-rat IgG (Invitrogen, Molecular Probe, A21208, 1:1,000), Cy5 AffiniPure Donkey Anti-Goat IgG (H+L) (Jackson, 705-175-147,1:300), Cy5 AffiniPure Donkey Anti-mouse IgG (H+L) (Jackson, 715-175-150,1:300), Cy5 AffiniPure Donkey Anti-rabbit IgG (H+L) (Jackson, 711-175-152,1:300).

## Real-time quantitative PCR

Vazyme FastPure Cell/Tissue Total RNA Isolation Kit was used to extract total RNA from organoids using. Approximately 1 μg of RNA was used to generate cDNA. Real-time quantitative PCR was performed using the Vazyme ChamQ Universal SYBR qPCR Master Mix. Primers used were as follows: PAX6 forward: 5′-TGGGCAGGTATTACGAGACTG -3′, reverse: 5′-ACTCCCGCTTATACTGGGCTA -3′; TBR1 forward: 5′-ATGGGCAGATGGTGGTTTTA-3′, reverse: 5′- GACGGCGATGAACTGAGTCT-3′; SATB2 forward: 5′-CCTCCTCCGACTGAA-GACAG-3′, reverse: 5′-TGGTCTGGGTACAGGCCTAC-3′; DLX2 forward: 5′-ACGCTCCC TATGGAACCAGTT-3′, reverse: 5′-TCCGAATTTCAGGCTCAAGGT-3′; NKX2.1 forward: 5′-CGACTCCGTTCTCAGTGTCTGA-3′, reverse: 5′-CCTCCATGCCCACTTTCTTG-3′.

## Single-cell RNA-sequencing library preparation and data analysis

We used the Neurosphere Dissociation Kit (Miltenyi, 130-095-943) to approach single-cell suspensions from cultured hStrOs. Six to 8 hStrOs were randomly selected from hES and hiPS cell line at the Day 110 to obtain a single-cell suspension. hStrOs were transported in a sterile culture dish with 10 ml 1× Dulbecco's Phosphate-Buffered Saline (DPBS; Thermo Fisher Scientific, 14190144) on ice to remove the residual tissue storage solution. Dissociated cells were washed with 1× DPBS containing 2% FBS. Cells were stained with 0.4% Trypan blue (Thermo Fisher Scientific, 14190144) to check the viability of Countess II Automated Cell Counter (Thermo Fisher Scientific).

Beads with the unique molecular identifier (UMI) and cell barcodes were loaded close to saturation so that each cell was paired with a bead in a Gel Beads-in-emulsion (GEM). After exposure to cell lysis buffer, polyadenylated RNA molecules hybridized into the beads. Beads were retrieved into a single tube for reverse transcription. On cDNA synthesis, each cDNA molecule was tagged on the 5′ end (the 3′ end of a messenger RNA transcript) with UMI and cell label indicating its cell of origin. Briefly, 10× beads were subject to second-strand cDNA synthesis, adaptor ligation, and universal amplification. Sequencing libraries were prepared using randomly interrupted whole-transcriptome amplification products to enrich the 3′ end of the transcripts linked with the cell barcode and UMI. All the remaining procedures, including the library construction, were performed according to the standard manufacturer's protocol (Chromium Single Cell 3′ v3). Sequencing libraries were quantified using a High Sensitivity DNA Chip (Agilent) on a Bioanalyzer 2100 and the Qubit High Sensitivity DNA Assay (Thermo Fisher Scientific). The libraries were sequenced on NovaSeq6000 (Illumina) using 2 × 150 chemistry.

Reads were processed using the Cell Ranger 4.0 pipeline with default and recommended parameters. Next, Gene-Barcode matrices were generated for each sample by counting UMIs

and filtering non-cell-associated barcodes. Finally, we generate a gene-barcode matrix containing the barcoded cells and gene expression counts. This output was then imported into the Seurat (v3.2.0) R toolkit for quality control and downstream analysis of our single-cell RNA-seq data. All functions were run with default parameters unless specified otherwise. We first filtered the matrices to exclude low-quality cells using a standard panel of 3 quality criteria: (1) number of detected transcripts (number of UMIs); (2) detected genes; and (3) percent of reads mapping to mitochondrial genes ($\leq$10%). The expression of mitochondria genes was calculated using PercentageFeatureSet function of the seurat package. The normalized data (NormalizeData function in seurat package) was performed for extracting a subset of variable genes. Variable genes were identified while controlling for the strong relationship between variability and average expression. Next, we integrated data from different samples after identifying "anchors" between datasets using FindIntegrationAnchors and IntegrateData in the seurat package. Then, we performed principal component analysis (PCA) and reduced the data to the top 30 PCA components after scaled the data. We visualized the clusters on a 2D map produced with UMAP.

## RNA-sequencing library preparation

For RNA-seq, hStrOs were randomly collected after 60 days ($n$ = 10 organoids) and 80 days ($n$ = 16 organoids) cultured. Total RNA was extracted from the tissue using TRIzol Reagent (Plant RNA Purification Reagent for plant tissue) according to the manufacturer's instructions (Invitrogen), and genomic DNA was removed using DNase I (Takara). Then, RNA quality was determined by 2100 Bioanalyser (Agilent) and quantified using the ND-2000 (NanoDrop Technologies). Only a high-quality RNA sample (OD260/280 = 1.8~2.2, OD260/230 $\geq$ 2.0, RIN $\geq$ 6.5, 28S:18S $\geq$ 1.0, >1 μg) was used to construct a sequencing library. RNA-seq transcriptome library was prepared following TruSeqTM RNA sample preparation Kit from Illumina (San Diego, California, United States of America) using 1 μg of total RNA. Shortly, messenger RNA was first isolated according to the polyA selection method by oligo(dT) beads and then fragmented by fragmentation buffer. Secondly, double-stranded cDNA was synthesized using a SuperScript double-stranded cDNA synthesis kit (Invitrogen, California, USA) with random hexamer primers (Illumina). Then, the synthesized cDNA was subjected to end-repair, phosphorylation and "A" base addition according to Illumina's library construction protocol. Libraries were size selected for cDNA target fragments of 300 bp on 2% Low Range Ultra Agarose followed by PCR amplified using Phusion DNA polymerase (NEB) for 15 PCR cycles. After quantified by TBS380, paired-end RNA-seq sequencing library was sequenced with the Illumina HiSeq xten/NovaSeq 6000 sequencer (2 × 150 bp read length).

## EdU tracing

EdU (Click-iT EDU Alexa Flour High-Thoughput Imaging (HCS), Invitrogen A10072) working solution was prepared with neural differentiation media. To reduce the toxicity to hStrO and enable sparse cellular labeling, we incubate Day 35 hStrO with EdU overnight. Replace the EdU working solution with fresh neural differentiation media. hStrOs with EdU tracing were collected in Day 36 and Day 45. EdU Staining refer to instruction.

## Transplanting neurons into hStrOs

After dissociating the neurons, nearly 5,000 dissociated neurons were transplanted into the hStrOs using a modified needle and cultured on a spinning shaker for 24 h. After spinning for 24 h, the transplanted hStrOs were moved to a routine incubator. Allograft neurons in hStrOs

were recorded every 24 h by epifluorescence microscopy. For extended transplants, nearly 30,000 dissociated neurons were transplanted into the hStrOs.

## Sholl analysis

Hand monitoring images traced movements and activities of GFP neurons in the organoid. Cell traces were imported into ImageJ and analyzed using the built-in Sholl analysis feature. Concentric circles of 5 μm up to 200 μm from the soma center were used to quantify dendritic intersections.

## Quantification and statistical analysis

Cryosections collected were randomly selected. Unless specified, at least 3 nonadjacent sections in each organoid were selected for staining for each marker. Various cryosections are being quantified per organoid, and then to be averaged as 1 point. Unless specified, at least 4 fields of each section or coverslip were chosen randomly to record and count when collecting images. Unpaired *t* test and 1-way ANOVA were used to determine the statistical significance. $P < 0.05$ was determined as statistical significance. Sample sizes were estimated empirically. GraphPad Prism version 7.0.0 was used for statistical analyses.

## Supporting information

**S1 Fig. Generation of human striatal organoids. (A)** Measuring the perimeters of organoid from Day 0 to Day 44 i. Data, mean ± SD; organoids, *n* = 4. **(B)** Evaluating *PAX6*, *TBR1*, and *SATB2 transcription levels* by qPCR. Data, mean ± SD; Student *t* test; *, $P < 0.05$. **(C)** Evaluating *NKX2.1* transcription levels by qPCR. Values are plotted as the expression level (2-ΔCt) relative to the 0 μm group. Each data point corresponds to a pooled batch of 6 organoids. Data, mean ± SD. One-way ANOVA. *, $P < 0.05$, ns, no significant difference. The raw data underlying this figure can be found in the S2 Data.
(TIF)

**S2 Fig. hStrOs development mimics LGE development. (A)** The quantification of immunostaining with PAX6/SOX1 and OTX2/FOXG1 antibodies in the dissociated neurons from organoids revealed the telencephalic and the forebrain fates' induction in hStrOs (Day 20). A total of 10 organoids were used in dissociated cultured in each group. Data, mean ± SD. Scale bar, upper 50 μm; lower 40 μm. **(B)** Immunostaining with SOX2 and MAP2 antibodies revealed multiple VZ-like areas in hStrOs in cultured. The mean thickness of each VZ-like area (indicated with the arrows in the schematic diagram) was used for quantification (organoids, *n* = 8). Quantification of SOX2+ cells on Day 30, Day 45, and Day 80 (*n* = 6 organoids). Data, mean ± SD. One-way ANOVA, *, $P < 0.05$. Scale bar, 30 μm. **(C)** Immunostaining with DARPP32 and NEUN antibodies in hStrOs revealed the mature striatal MSNs in hStrOs on Day 60 and Day 80. Quantification of NEUN+ cells at Day 30, Day 45, and Day 80 (*n* = 6 organoids). Data, mean ± SD. One-way ANOVA, *, $P < 0.05$. Scale bar, 30 μm. **(D)** UMAP visualization of the resolved scRNA-seq data of hStrOs. Histogram showing the percentage of cells in each cell type belonging to 2 cell lines in hStrOs (ESC-H9: green; iPSC-8-12: red). **(E)** Gene reads in each cluster. **(F)** Schematic representation of cluster annotation. **(G–I)** UMAP visualization of expression of selected genes in the hStrOs scRNA-seq data at Day 110 of in vitro differentiation (*n* = 26,534 cells from hESC-H9 and hiPSCs-8-12). **(J)** Bulk RNA-seq and scRNA-seq from hStrOs (Day 60, Day 80, and Day 110) mapped to the Brain Span human brain dataset (PCW8, 9, 12, 13, and 16) by using VoxHunt. The raw data underlying this figure

can be found in the S2 Data.
(TIF)

**S3 Fig. Identification of regionalization in developing hStrO. (A)** The representative images of the hStrOs were collected on Day 25, and hCOs were collected on Day 45. Scale bar, 1 mm; insert, 100 μm. **(B)** Immunostainings for Ki67 and SOX2 antibodies revealed the expression of SOX2 and Ki67 in LGE units. Scale bar, 50 μm. **(C–E)** Immunostainings for Ki67 and MAP2 antibodies revealed progressive regionalization in Day 30 and 60 hStrOs. Arrows in C showed LGE units on Day 30 hStrO. Dashed lines marked the supposed region: (1) **LGE units**; (2) **Region ii**; (3) **Region i**; (4) **Necrotic areas**. Scale bar, C, above, 100 μm, down 150 μm; D, 30 μm; E, 20 μm.
(TIF)

**S4 Fig. Identification of regionalization in developing hStrO. (A)** Immunostainings for MASH1 and GABA antibodies revealed the distribution patterns of LGE progenitor cells on Day 45, Day 60, and Day 80 hStrOs. Arrows showed Region i (white) and Region ii (yellow). Scale bar, 100 μm. **(B–D)** Immunostainings for MASH1 and GABA antibodies revealed the distribution patterns of LGE progenitor cells in hStrOs. The LGE progenitor cells in hStrOs spatially separated from the mature neurons. Arrows showed the region that was absent from MASH1+ progenitor cells. Scale bar, B, 50 μm; C, 30 μm; D, 50 μm.
(TIF)

**S5 Fig. Regionalization in hStrO mimics compartment-like zones in striatum. (A, B)** Immunostaining with FOXP2, DARPP32, and MAP2 antibodies revealed the compartmentalization in the 22 W fetal striatum. Arrows showed mosaics embedded within the striatum formed by DARPP32+ or FOXP2+ patches. Scale bar, 300 μm; insert, 50 μm. **(C, D)** Immunostaining with FOXP2, DARPP32, and MAP2 antibodies revealed high expression of DARPP32 and FOXP2 in Region i of Day 45 hStrO. Arrows and dash line marked Region i. Scale bar, 200 μm.
(TIF)

**S6 Fig. Self-organized regionalization prevails in hStrO. (A, B)** Bright-field images showed that 89% of organoids (31/35) have a significant hyaline patch in a random batch of hStrO. Scale bar, 1mm. **(C, D)** Immunostaining with FOXP2 revealed that 79% of organoids (19/24) showed a transparent FOXP2 enriched Region i in random batches of hStrOs on Day 45. Scale bar, 1 mm. **(E)** Immunostaining with FOXP2 and MAP2 antibodies revealed the matrix striosome-like zones of organoids (Pur: 0.65, 0.7, and 0.75 μm), which is not observed in the organoid culture without Pur. Scale bar, 100 μm.
(TIF)

**S7 Fig. Self-organized regionalization in long-term cultured hStrO. (A)** Representative images showing 4 regions divided by Ki67 and FOXP2 expression in hStrOs. Scale bar, 30 μm. **(B)** Immunostaining of NeuN and DARPP32 antibodies revealed NEUN+DARPP32+ MSN in Mature regions i and ii on Day 45 to Day 80. Scale bar, 100 μm. **(C)** Immunostainings for Bassoon, PSD95, and Tuj1 antibodies reveal the existence of pre- and postsynaptic proteins in hStrOs after 45, 60, 80, and 110 days of culture. Scale bar, 5 μm. **(D)** Quantification of the number of Bassoon+ presynaptic and PSD95+ postsynaptic proteins in Mature region i and ii of hStrOs. Value on Y-axis = the counts of bassoon+ or PSD95+ puncta/volumes of Tuj1 + fibers rendered in IMARIS ($n$ = more than 10 clear Tuj1+ fibers rendered in IMARIS). One-way ANOVA, *, $P < 0.05$; ns, nonsignificant. The raw data underlying this figure can be found

in the S2 Data.
(TIF)

**S8 Fig. Typical rosette participating in hStrO regionalization.** Immunostainings for MEIS1/2, MASH1, and DARPP32 reveal typical neural tubes contributing to Region i and LGE units in Day 45 hStrO. Arrows showed the possible migrating routes of progenitors from the individual rosette to participate in the hStrO regionalization. Scale bar, 150 μm; insert, 50 μm.
(TIF)

**S9 Fig. Expression of FOXP2, CALB1, and CALB2 in developing hStrO. (A, B)** Immunostainings for CALB1, CALB2, and FOXP2 antibodies on Day 45 and Day 100 hStrOs. Images showed typical CALB1/CALB2 expression in Region i/ii. Co-expressed of GABA and MEIS1/2 identified the striatal fates. Scale bar, A, 20 μm; B, 30 μm.
(TIF)

**S10 Fig. Distribution of mature neurons in developing hStrO. (A, B)** Immunostainings for MAP2, SOX2, and FOXP2 antibodies revealed the mature neurons' distribution in Region i and Region ii of Day 60 hStrOs. White arrows showed SOX2+ cells, and yellow arrows showed SOX2-MAP2+ mature neurons. The left panels are the magnified region from the boxed area, shown as 2 or spliced channels. Scale bar, 40 μm; insert, 15 μm, individual cell, 2 μm. **(C, D)** Immunostainings for MAP2, SOX2, and FOXP2 antibodies revealed mature neuronal distribution in different slices of the same organoids. From left to right, Region i/ii showed an increase in mature neurons. Scale bar, C, 50 μm; D, 40 μm.
(TIF)

**S11 Fig. EdU tracing in hStrO. (A)** Schematic of EdU tracing in hStrO. **(B)** EdU tracing showed EdU+ cells were predominantly identified in the rosette while excluded from the rosette after 10 days. Scale bar, 30 μm. **(C)** Immunostainings for MAP2 and FOXP2 antibodies revealed EdU+ cells were found in both LGE units and Region i/ii. Scale bar, 30 μm.
(TIF)

**S12 Fig. Fused organoids reconstructed the cortical or midbrain projections targeting the regions of striatal organoids. (A)** Immunostaining for PAX6 and TBR2 antibodies on Day 45 hCOs. Scale bar, 50 μm. **(B)** Immunostaining for TBR1, CTIP2, and SATB2 antibodies in hCOs on Day 45. Scale bar, 40 μm. **(C)** Immunostaining images for FOXA2 and TH antibodies on Day 50 hMOs. The lower panel is the single ortho view of Z-stack images. Scale bar in the upper panel, 30 μm; in the lower panel, 30 μm. **(D)** Images of hStr-StrOs at dpf 13. The lower panels are the magnified images. Red arrows showed GFP+ migrating cells from hStrO. Scale bar, 100 μm. **(E, F)** Immunostaining for GFP and GABA antibodies revealed the intermingled GFP+ axons with GABA+ neurons in hC-StrOs and hM-StrOs at dpf 20. The left panels show the tomography of the boxed region rendered by Imaris. Scale bar, 20 μm. **(G, H)** Immunostaining for GFP, Bassoon, and PSD95 antibodies revealed contact between a pre- and postsynapse in hC-StrOs and hM-StrOs at dpf 20. Arrows showed the contact of Bassoon + presynapses with PSD95+ postsynapses on GFP+ axons. Scale bar, G, 30 μm; H, 20 μm. **(I, J)** Immunostaining for GFP, GABA, and PSD95 antibodies revealed GFP+ axons projecting towards PSD95+ puncta on GABA neurons in hC-StrOs and hM-StrOs at dpf 20. Arrows showed the contact of GABA+ neurons with PSD95+ puncta in GFP+ axons. The right 3 panels show the relationship between the 2 signals. Scale bar, 30 μm.
(TIF)

**S13 Fig. Migration and synaptogenesis of HD cells graft in hStrO. (A)** Fluorescent image of hStrOs reveals that neurons exhibit migratory-like behavior after coculture. The right panel

shows the migration trajectory plotted after overlapping the fluorescence photos at a different time (72, 96, 120, and 144 h). Scale bar, 200 μm. **(B)** Immunostaining for bassoon, PSD95, Tuj1, and GFP antibodies reveals the anatomical integration between GFP+ transplanted neurons and hStrO. The boxed region is the magnified region in the middle panel; the rendered tomography of fluorescent signals is in the right panel. dat, days after transplanted. Scale bar, 30 μm.
(TIF)

**S1 Table. Antibodies for immunostaining with their purpose.**
(PDF)

**S2 Table. Comparison of protocols for striatal neuron differentiation from human pluripotent stem cells.**
(PDF)

**S1 Data. The raw data used in Figs 1–6 are included in S1 Data.** Excel spreadsheet containing, in separate sheets, the underlying raw data for figure panels Figs 1B, 1D, 1G, 1I, 1K, 2B, 2D, 2E, 2F, 2K, 3D, 3E, 3H, 4C, 4D, 5F, 5I, 5J, 6B, 6D, 6F, 6G and 6I.
(XLSX)

**S2 Data. The raw data used in S1, S2 and S7 Figs are included in S2 Data.** Excel spreadsheet containing, in separate sheets, the underlying raw data for figure panels S1A–S1C, S2A–S2E, S2G, S2J and S7D Figs.
(XLSX)

# Acknowledgments

The authors thank the HD family in Inner Mongolia in China for making this project possible and Dr. Ren Yulei for TEM scanning. Also, we thank Shanghai Majorbio Bio-pharm Technology for providing RNA sequencing service and single-cell sequencing. Dr. Dong Yachen, Zhu Tong, and He Xiaolong from Majorbio Bio-pharm Technology analyzed RNA and single-cell sequencing data and visualized all sequencing data.

# Author Contributions

**Conceptualization:** Xinyu Chen, Hexige Saiyin, Lixiang Ma.

**Data curation:** Xinyu Chen, Yang Liu, Yuqi Wang, Xuan Li, Rong Ji.

**Formal analysis:** Xinyu Chen, Yang Liu.

**Funding acquisition:** Lixiang Ma.

**Methodology:** Xinyu Chen, Hexige Saiyin.

**Project administration:** Hexige Saiyin, Lixiang Ma.

**Resources:** Hexige Saiyin, Lixiang Ma.

**Software:** Hexige Saiyin.

**Supervision:** Hexige Saiyin, Lixiang Ma.

**Visualization:** Xinyu Chen.

**Writing – original draft:** Xinyu Chen.

**Writing – review & editing:** Hexige Saiyin, Lixiang Ma.

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
