## [Editor Report · Decision Letter 0]

3 Mar 2022

Dear Dr Ma, 

Thank you for submitting your manuscript entitled "Human striatal organoids derived from pluripotent stem cells recapitulate striatal development and compartments" for consideration as a Research Article by PLOS Biology.

Your manuscript has now been evaluated by the PLOS Biology editorial staff as well as by an academic editor with relevant expertise and I am writing to let you know that we would like to send your submission out for external peer review.

Once your full submission is complete, your paper will undergo a series of checks in preparation for peer review. Once your manuscript has passed the checks it will be sent out for review. To provide the metadata for your submission, please Login to Editorial Manager (https://www.editorialmanager.com/pbiology) within two working days, i.e. by Mar 07 2022 11:59PM.

If your manuscript has been previously reviewed at another journal, PLOS Biology is willing to work with those reviews in order to avoid re-starting the process. Submission of the previous reviews is entirely optional and our ability to use them effectively will depend on the willingness of the previous journal to confirm the content of the reports and share the reviewer identities. Please note that we reserve the right to invite additional reviewers if we consider that additional/independent reviewers are needed, although we aim to avoid this as far as possible. In our experience, working with previous reviews does save time. 

If you would like to send previous reviewer reports to us, please email me at ialvarez-garcia@plos.org to let me know, including the name of the previous journal and the manuscript ID the study was given, as well as attaching a point-by-point response to reviewers that details how you have or plan to address the reviewers' concerns. 

Given the disruptions resulting from the ongoing COVID-19 pandemic, please expect some delays in the editorial process. We apologise in advance for any inconvenience caused and will do our best to minimize impact as far as possible.

Kind regards,

Ines

--

Ines Alvarez-Garcia, PhD

Senior Editor

PLOS Biology

---

## [Decision Letter · Decision Letter 1]

9 May 2022

Dear Dr Ma,

Thank you for your patience while your manuscript entitled "Human striatal organoids derived from pluripotent stem cells recapitulate striatal development and compartments" was peer-reviewed at PLOS Biology and please accept my apologies for the delay in providing you with our decision. The manuscript has now been evaluated by the PLOS Biology editors, an Academic Editor with relevant expertise, and by three independent reviewers. 

The reviews are attached below. As you will see, the reviewers find the conclusions potentially interesting, however they also raise several issues regarding the structure, interpretation and the way some of the experiments were performed. Reviewer 1 asks for some clarifications, to discuss the strength of the method over existing ones and suggests that using an electrophysiological approach or calcium imaging would be better to show the maturation of striatal neurons. Reviewers 2 and 3 are more critical and find the manuscript difficult to follow due to the structure and writing. They think that the compartmentalization needs further work, and that the projection and disease data should be fleshed out.

In light of the reviews, we would like to invite you to revise the work to thoroughly address the reviewers' reports. Given the extent of revision needed, we cannot make a decision about publication until we have seen the revised manuscript and your response to the reviewers' comments. Your revised manuscript is likely to be sent for further evaluation by all or a subset of the reviewers.

**IMPORTANT - SUBMITTING YOUR REVISION**

3. Resubmission Checklist

a) *PLOS Data Policy*

b) *Published Peer Review*

c) *Blurb*

Please also provide a blurb which (if accepted) will be included in our weekly and monthly Electronic Table of Contents, sent out to readers of PLOS Biology, and may be used to promote your article in social media. The blurb should be about 30-40 words long and is subject to editorial changes. It should, without exaggeration, entice people to read your manuscript. It should not be redundant with the title and should not contain acronyms or abbreviations. For examples, view our author guidelines: https://journals.plos.org/plosbiology/s/revising-your-manuscript#loc-blurb

Sincerely,

Ines

--

Ines Alvarez-Garcia, PhD

Senior Editor

PLOS Biology

Reviewers' comments

Rev. 1:

In this manuscript, Chen et al. used human pluripotent stem cells to generate human striatal organoids (hStrOs) that recapitulates the development of fetal striatum. Although hStrOs were generated and reported by other groups before, here authors comprehensively investigated the cellular composition and developmental time course of cellular and molecular changes through different stages of hStrO development with immunohistochemical and transcriptomics approaches. The method authors developed induced the development of distinct striatal compartments in a single organoid including striosomes and matrix. Also, using organoid fusions, they have identified neural migration/axon guidance differences between striatum and different brain regions including the cortex and the midbrain. Finally, they have transplanted Huntington's disease patient-derived striatal neurons into the day 45 hStrOs and showed successful engraftment of these neurons with healthy neurons that can be used as a model to investigate functional integration of donor neurons into the host.

Overall, the authors developed a new approach to generate striatum organoids with distinct striatal compartment, they had extensively characterized the cellular and molecular features, and they have shown that their hStrOs can be used to model cellular interactions with other brain regions, and these organoids can be used as a host for donor-host integration studies. Moreover, the manuscript is well written, and the figures are well-organized. There are only a couple of questions/suggestions which may improve the conclusions of the manuscript.

1. The striatum develops from the lateral ganglionic eminence (LGE), which is located at the ventral part of the developing forebrain. This region is known to express dorsal forebrain marker Pax6 at low levels, which is also supported by authors' data in Fig. 1D-F and Fig. S1B ( <1% Pax6-positive cells). However, following these results, in Fig S2A-B, they showed that close to 90% of cells dissociated from hStrOs are Pax6-positive. Is this a discrepancy in their data? Is it due to the samples taken at different days in culture? Authors need to clarify this either in the result section or discuss it in the discussion section.

2. In the introduction section, line 8, authors used the terms 'self-patterning' and 'unself-patterning' possibly referring to the generation of region-specific vs cerebral organoids through guided vs unguided patterning protocols. Please correct these terms as "guided and unguided methods" since unself-patterning is not really a term to describe unguided methods.

3. In general, authors never discussed in their manuscript the strength of their method to generate hStrOs over the previous methods. It would be nice to add a couple of sentence regarding this at least in the Discussion.

4. As they mentioned in the Discussion, rather than only showing pre- and post-synaptic marker staining data in fused organoids, an electrophysiological approach or calcium imaging would better show the maturation of striatal neurons with or without input from cortical/midbrain neurons.

5. Although the injection of patient-derived neurons and subsequent migratory characteristics of them in hStrOs show that hStrOs can serve as a host for these neurons, I do not totally grasp the importance of this data and how it makes their hStrOs a better model. This is a process which can possibly occur in any type of organoids. Please discuss the importance of this experiment and what it proves.

Rev. 2:

In this manuscript, the authors are characterizing human iPSC and stem cell derived organoids (in isolation and fused, relative to region) to identify conditions for improved development of striatal organoids which better represent the cell types and physical compartmentalization of the LGE-derived striatum. The main variable is concentration of puromorphine, which they confirm is required for "striatal" development from iPSCs. This reviewer found it extremely difficult to read and follow this manuscript due to jumping around between markers, supposed regions of the organoids, overuse of supplementary data for key points (e.g. regionalization) and the number of small and difficult to interpret figures some of which are not labeled with what they are stained. Also, this reviewer is very sympathetic to the fact that English is not everybody's first language, but the grammar errors herein actually sometimes make the text difficult to understand, and it would have been helpful to have the language edited prior to review. The manuscript would also really benefit by an initial Table 1 listing the markers that are used and what they are purported to represent. That being said, the main conclusions are that the sonic hedgehog pathway is required for regionalization, i.e. that puromorphine was required, and that compartment and regional specific synapses are visualized in fused organoids.

There is a theoretical problem in claiming compartmentalization, which appears to be primarily based on visual distinctions between what is first called "hyaline patches" not seen in cortical organoids, and then by regionalization noted by early and late LGE markers and two regions which differed in cell density. There is a problem here in that DARPP-32 is used as the definitive marker of MSNs in all regions, so it is not clear what is really distinguishing the regions neurochemically. Single cell RNA seq is not adequate for this purpose as the clusters are not spatially defined. More problematic, and apologies if this was missed, all DARPP-32+ cells should be GABAergic if they are to be defined as MSNs, and several figures, particularly in Figure 2, do not appear to show that this is the case. Of course it goes without saying also that all DARPP-32+ and CTIP2 (better referred to as Bcl11b) should be NeuN+, and the lack of double label with DARPP32 and MAP2 is also confusing.

Part of the problem lies in the fact that DARPP-32 is indeed initially a marker of striosome cells in vivo, but then its expression becomes generalized, and although the authors refer to the fact that striosome markers change with development, they make no effort to explain to the reader what phase of development they think they are modeling or characterizing. Within the striosome regions, the authors then should demonstrate double label of DARPP-32/FoxP2. Also if there are specific striosome regions, then presumably the authors are thinking that the remaining striatal-like portion of the organoid represents matrix? but again this is characterized only by scRNA seq. It should also be noted that in early striatal development, calbindin localizes more to striosomes than matrix, again making interpretation here difficult vis a vis what developmental period the authors think each time point represents.

It is very difficult to comment specifically on the experiments with "HD cells" due to the lack of characterization of these cells. Likewise, the specificity of connections in the fused organoids is difficult to validate without the same characterization re dopaminergic neurons, etc, as was performed in isolated organoids.

Rev. 3:

The manuscript by Chen et al titled "Human striatal organoids derived from pluripotent stem cells recapitulate striatal development and compartments" describes a method to generate striatal organoids containing DARPP32 cells and regions that appear to resemble the striatum striosome and matrix compartments. In addition, the authors describe striatal assembloids that show projections that appear to be similar to projection patterns in vivo. Because striatal organoids and assembloids have already been described by a different group (Miura et al 2020), the main novelty of the work is the formation of discrete striosome and matrix compartments. In vivo these compartments are characterized by distinct histological markers and afferent and efferent connectivity. In the manuscript, however, the characterization of these compartments in organoids was not thorough or convincing. More specific concerns are described below:

1) While this reviewer appreciates the authors' effort to quantify a big number of striatal markers by immunohistochemistry, there are some issues with the quantification:

- The numbers in Figures 1 and 2 do not seem to match. For example, Figure 1 shows that ~70% of DAPI at d45 are DARPP32+, yet Figure 2 and S2 show that ~60% of DAPI at this age are SOX2. I would not expect DARPP32 cells to be SOX2+. In another example, Figure 2 shows that ~80% of DARPP32 cells are NeuN+, yet NeuN cells only account for 5% of all DAPI. Are most DARPP32 cells NeuN negative then?

- It is unclear what the points in each graph represent. Figure legends say "n = 4-6 organoids", yet there are a lot more points in each graph. If various cryosections are being quantified per organoid, then these need to be averaged and each point should represent an organoid and statistics should be performed with organoids as n.

2) The authors make a claim about the position of cells within an organoid; however, this should be quantified. For example, although one image shows MASH1 cells in "SVZ", and CTIP2 cells in "MZ", Figure 2F shows CTIP2 cells colocalizing with MASH1 in the SVZ.

3) Parts of figure 1 and 2 seem to be repeating themselves, they can be consolidated and reorganized (currently the flow is not very clear and jumps from progenitors to neurons and back for no clear reason). Either have the single cell as a separate figure, or have a figure for progenitors and another for neurons

4) The authors should explain the rationale of doing most of the characterization of the organoids at d45 yet performing scRNAseq at d110.

5) Quantification data for MASH1 at d45 in Figures 2E and 2N looks very similar. Please double check that the same data was not used for both graphs.

6) The rationale behind the characterization of striosome and matrix compartments was not clear, and further markers and quantification would be needed to claim these two compartments are formed in organoids. For example:

- In figure 3B - why are KI67 negative areas called mature regions? They are not co-stained with any other marker that would suggest they are "mature"

- Moreover, it looks like FOXP2 cells in "mature region I" are MAP2 negative (Fig 3H, I). Is that expected?

- In Figure 3D - Why are DARPP32 regions MAP2 negative? Based on previous figures most DARPP32 cells should be MAP2 positive

- In the scRNAseq dataset, while there is expression of striosome/matrix genes, they are scattered and they don't seem to be colocalized. Eg. ID4 and CALB1 are not expressed in the same cells and OPRM1 and SEMA5B are not expressed in same cells

- How many organoids was the regionalization seen in? (Fig 3F,G) This is not specified in the legend.

- Striosome and matrix compartments are characterized by the expression of a number of markers. Have the authors checked expression of DRD1 and DRD2, GAD1, SST, TAC1, PENK, etc? It would be important to show more markers for these regions

7) Although it is in principle an interesting idea to graft HD cells, it is unclear what the purpose of this experiment was. Why not use control GFP+ cells?

Minor

- In introduction: unself-patterning - there is not such a thing as unself-patterning

- In introduction: "spatially self-organized and structurally and functionally recapitulate the neuronal activities of the human brain" -> tone down

- DARPP32 staining organoids in Figure 1G is not convincing - better picture would be ideal. Figure S2E has a better staining

- DLX2 is also an MGE marker yet it is being used as an exclusive LGE marker

- Looking at the figures and figure legends, it is unclear when 2D vs 3D is being shown/quantified. Please specify clearly

- SOX2 staining looks unspecific. Note that it appears to be expressed in VZ and outside VZ (where MAP2 neurons are). I would not expect MAP2 neurons to be SOX2+

- Clarify what Fig 2E is showing. The text implies that it is quantifying the position of GSX2/MASH1 and CTIP2 cells within the organoid, yet the figure legend indicates it is showing quantification of expression. "We found that cells labeled by different markers have their own preferences in the location: GSH2+ cells, mostly located in VZ-like zones (33.03 ± 11.84%); MASH1+ cells, sprinkled closed to the VZ-like zones (23.8 ± 9.977%); CTIP2+ cells, largely excluded from VZ-like zones (42.62 ± 12.83%)"

- Gene names in single cell analysis should be italic

- Specify what the colors represent in Fig 3L

- Images in 4B do not clearly depict projections and or migrating neurons. Except for one of the arrows where one can see a projection, it is hard to see what the rest of the arrows are pointing at

- Significance is not shown in Fig 4I

- "it is rational to consider these results as reminiscent of the striosomal MSNs receiving preferential inputs from a deep cortical layer in vivo" - authors do not show projections being from deep or superficial cortical neurons.

---

## [Editor Report · Decision Letter 2]

17 Aug 2022

Dear Dr Ma,

Thank you for submitting your revised manuscript entitled "Human striatal organoids derived from pluripotent stem cells recapitulate striatal development and compartments" for publication as a Research Article at PLOS Biology. This revised version of your manuscript has been evaluated by the PLOS Biology editors and the Academic Editor.

Based on our Academic Editor's assessment of your revision, we are likely to accept this manuscript for publication, provided you satisfactorily address the data and other policy-related requests stated below.

In addition, we think that your manuscript would fit better as a Methods and Resources paper, thus we would like to consider it for publication in that format. When you submit the final version, please select that article type from the dropdown menu.

We expect to receive your revised manuscript within two weeks. 

*Published Peer Review History*

*Press*

Sincerely,

Ines

--

Ines Alvarez-Garcia, PhD

Senior Editor

PLOS Biology

Fig. 1B, D, G, I, K; Fig. 2B, D, E, F, K; Fig. 3D, E, H; Fig. 4C, D; Fig. 5F, I, J; Fig. 6B, D, F, I; Fig. S1A-C; Fig. S2A-E, G-J; Fig. S3 and Fig. S7D

BLURB

Please also provide a blurb which (if accepted) will be included in our weekly and monthly Electronic Table of Contents, sent out to readers of PLOS Biology, and may be used to promote your article in social media. The blurb should be about 30-40 words long and is subject to editorial changes. It should, without exaggeration, entice people to read your manuscript. It should not be redundant with the title and should not contain acronyms or abbreviations. For examples, view our author guidelines: https://journals.plos.org/plosbiology/s/revising-your-manuscript#loc-blurb

---

## [Editor Report · Decision Letter 3]

28 Sep 2022

Dear Dr Ma,

Thank you for your patience while we considered your revised manuscript entitled "Human striatal organoids derived from pluripotent stem cells recapitulate striatal development and compartments" for publication as a Methods and Resources at PLOS Biology. Please also accept my apologies for the delay with your manuscript.

While we are now satisfied with the medatada, Reviewer 2 contacted us belatedly expressing remaining concerns regarding the inaccuracy of some of the statements on MSNs and DARPP032 included in the manuscript. For example, some sentences such as “self-patterning and unself-patterning” and an explanation that DARPP-32 exists in immature neurons, which require a better explanation to compensate for the lack of double-label with MAP2. The reviewer also mentioned that the co-localization of DARPP-32 with GABA, or lack thereof, has not been addressed and that Calretinin/CALB2 immunoreactive fibers are enriched in striosomes, not neurons. After consulting with the Academic Editor, we would like you to adjust your interpretation to point out that any one marker is never enough to absolutely identify a cell type, but rather multiple markers are needed. Please tone down some of the statements to mention that while the data suggest these cells have MSNs, there may also be other identities that also exhibit some of those markers.

We expect to receive your revised manuscript within two weeks. 

- a cover letter that should detail your responses to any editorial requests, if applicable,

Sincerely,

Ines

--

Ines Alvarez-Garcia, PhD

Senior Editor

PLOS Biology

---

## [Editor Report · Decision Letter 4]

5 Oct 2022

Dear Dr Ma,

Thank you for the submission of your revised Methods and Resources entitled "Human striatal organoids derived from pluripotent stem cells recapitulate striatal development and compartments" for publication in PLOS Biology. On behalf of my colleagues and the Academic Editor, Madeline Lancaster, I am happy to say that we can in principle accept your manuscript for publication, provided you address any remaining formatting and reporting issues. These will be detailed in an email you should receive within 2-3 business days from our colleagues in the journal operations team; no action is required from you until then. Please note that we will not be able to formally accept your manuscript and schedule it for publication until you have completed any requested changes.

PRESS

Sincerely, 

Ines

--

Ines Alvarez-Garcia, PhD

Senior Editor

PLOS Biology
